# Prediction-Powered Ranking of Large Language Models

**Ivi Chatzi**
Max Planck Institute for
Software Systems
Kaiserslautern, Germany
ichatzi@mpi-sws.org

**Eleni Straitouri**
Max Planck Institute for
Software Systems
Kaiserslautern, Germany
estraitouri@mpi-sws.org

**Suhas Thejaswi**
Max Planck Institute for
Software Systems
Kaiserslautern, Germany
thejaswi@mpi-sws.org

**Manuel Gomez Rodriguez**
Max Planck Institute for
Software Systems
Kaiserslautern, Germany
manuelgr@mpi-sws.org

## Abstract

Large language models are often ranked according to their level of alignment with human preferences—a model is better than other models if its outputs are more frequently preferred by humans. One of the popular ways to elicit human preferences utilizes pairwise comparisons between the outputs provided by different models to the same inputs. However, since gathering pairwise comparisons by humans is costly and time-consuming, it has become a common practice to gather pairwise comparisons by a strong large language model—a model strongly aligned with human preferences. Surprisingly, practitioners cannot currently measure the uncertainty that any mismatch between human and model preferences may introduce in the constructed rankings. In this work, we develop a statistical framework to bridge this gap. Given a (small) set of pairwise comparisons by humans and a large set of pairwise comparisons by a model, our framework provides a rank-set—a set of possible ranking positions—for each of the models under comparison. Moreover, it guarantees that, with a probability greater than or equal to a user-specified value, the rank-sets cover the true ranking consistent with the distribution of human pairwise preferences asymptotically. Using pairwise comparisons made by humans in the LMSYS Chatbot Arena platform and pairwise comparisons made by three strong large language models, we empirically demonstrate the effectivity of our framework and show that the rank-sets constructed using only pairwise comparisons by the strong large language models are often inconsistent with (the distribution of) human pairwise preferences.

## 1 Introduction

During the last years, large language models (LLMs) have shown a remarkable ability to generate and understand general-purpose language [1]. As a result, there has been an increasing excitement in their potential to help humans solve a variety of open-ended, complex tasks across many application domains such as coding [2], healthcare [3] and scientific discovery [4], to name a few. However, evaluating and comparing the performance of different LLMs has become very challenging [5]. The main reason is that, in contrast to traditional machine learning models, LLMs can solve a large number of different tasks and, in many of these tasks, there is not a unique, structured solution. As a

38th Conference on Neural Information Processing Systems (NeurIPS 2024).

consequence, there has been a paradigm shift towards evaluating their performance according to their level of alignment with human preferences—a model is better than other models if its outputs are more frequently preferred by humans [6–10].

One of the most popular paradigms to rank a set of LLMs according to their level of alignment with human preferences utilizes pairwise comparisons [10–17]. Under this paradigm, each pairwise comparison comprises the outputs of two different models picked uniformly at random to an input sampled from a given distribution of inputs. Moreover, the pairwise comparisons are used to rank the models with a variety of methods such as the Elo rating [18–22], the Bradley-Terry model [10, 17, 23] or the win-rate [12, 17, 23]. While it is widely agreed that, given a sufficiently large set of pairwise comparisons, higher (lower) ranking under this paradigm corresponds to better (worse) human alignment, there have also been increasing concerns that this paradigm is too costly and time-consuming to be practical, especially given the pace at which models are updated and new models are developed.

To lower the cost and increase the efficiency of ranking from pairwise comparisons, it has become a common practice to ask a strong LLM—a model known to strongly align with human preferences—to perform pairwise comparisons [24–33]. The rationale is that, if a model strongly aligns with human preferences, then, the distributions of pairwise comparisons by the model and by the human should in principle match [24, 27, 34]. Worryingly, there are multiple lines of evidence, including our experimental findings in Figure 3, showing that the rankings constructed using pairwise comparisons made by a strong LLM are sometimes different to those constructed using pairwise comparisons by humans [12, 14–16, 19, 35, 36], questioning the rationale above. In this work, we introduce a statistical framework to measure the uncertainty in the rankings constructed using pairwise comparisons made by a model, which may be introduced by a mismatch between human and model preferences or by the fact that we use a finite number of pairwise comparisons.

**Our contributions.** Our framework measures uncertainty using rank-sets—sets of possible ranking positions that each model can take. If the rank-set of a model is large (small), it means that there is high (low) uncertainty in the ranking position of the model. To construct the rank-sets, our framework first leverages a (small) set of pairwise comparisons by humans and a large set of pairwise comparisons by a strong LLM to create a confidence ellipsoid. By using prediction-powered inference [37–39], this confidence ellipsoid is guaranteed to contain the vector of (true) probabilities that each model is preferred over others by humans—the win-rates—with a user-specified coverage probability $1 - \alpha$. Then, it uses the distance between this ellipsoid and the hyperplanes under which pairs of models have the same probability values of being preferred over others to efficiently construct the rank-sets. Importantly, we can show that, with probability greater than or equal to $1 - \alpha$, the constructed rank-sets are guaranteed to cover the ranking consistent with the (true) probability that each model is preferred over others by humans asymptotically. Moreover, our framework does not make any assumptions on the distribution of human preferences nor about the degree of alignment between pairwise preferences of humans and the strong LLM. Experiments on pairwise comparisons made by humans in the LMSYS Chatbot Arena platform [28] and pairwise comparisons made by three strong LLMs, namely GPT 3.5, Claude 3 and GPT 4, empirically demonstrate that the rank-sets constructed using our framework are more likely to cover the true ranking consistent with (the distribution of) human pairwise preferences than the rank-sets constructed using only pairwise comparisons made by the strong LLMs. An open-source implementation of our methodology as well as the data on pairwise preferences of strong LLMs used in our experiments are available at https://github.com/Networks-Learning/prediction-powered-ranking.

**Further related work.** Our work builds upon recent work on prediction-powered inference, ranking under uncertainty, and ranking of LLMs.

Prediction-powered inference [37–39] is a recently introduced statistical framework to obtain valid $p$-values and confidence intervals about a population-level quantity such as the mean outcome or a regression coefficient using a small labeled dataset and a large unlabeled dataset, whose labels are imputed using a black-box machine learning model. However, our work is the first to use prediction-powered inference (as a subroutine) to construct rank-sets with coverage guarantees. In this context, it is worth acknowledging that a very recent work by Saad-Falcon et al. [40] has used prediction-powered inference to construct (single) rankings, rather than rank-sets. However, their rankings do not enjoy coverage guarantees with respect to the true ranking consistent with (the distribution of) the human preferences. Moreover, an independent, concurrent work by Boyeau

et al. [23] has also used prediction-powered inference to construct (single) rankings based on the estimated coefficients of a Bradley-Terry model. However, the estimated coefficients come with large, overlapping confidence intervals, which would have led to uninformative rank-sets, had the authors used them to construct rank-sets.

The vast majority of the literature on ranking under uncertainty has focused on confidence intervals for individual ranking positions [41–47]. Only recently, a paucity of work has focused on joint measures of uncertainty for rankings [48–51]. Similarly as in our work, this line of work also seeks to construct rank-sets with coverage guarantees. However, in contrast to our work, it estimates the quality metric (in our work, the probability that an LLM is preferred over others) and the confidence intervals separately for each of the items (in our work, LLMs) using independent samples. As a consequence, it needs to perform multiple comparison correction to create the rank-sets.

In recent years, there has also been a flurry of work on ranking LLMs using benchmark datasets with manually hand-crafted inputs and ground-truth outputs [52–58]. However, it has become increasingly clear that oftentimes rankings derived from benchmark datasets do not correlate well with rankings derived from human preferences—an improved ranking position in the former does not lead to an improved ranking position in the latter [12–14, 17, 26]. Within the literature on ranking LLMs from pairwise comparisons, most studies use the Elo rating system [18–22], originally introduced for chess tournaments [59]. However, Elo-based rankings are sensitive to the order of pairwise comparisons, as newer comparisons have more weight than older ones, which leads to unstable rankings [15]. To address this limitation, several studies have instead used the Bradley-Terry model [10, 17, 23], which weighs pairwise comparisons equally regardless of their order. Nevertheless, both the Elo rating system and the Bradley-Terry model have faced criticism, as pairwise comparisons often fail to satisfy the fundamental axiom of transitivity, upon which both approaches rely [15, 60], Recently, several studies have used the win-rate [12, 17, 23], which weighs comparisons equally regardless of their order and does not require the transitivity assumption, but requires humans to make pairwise comparisons between every pair of models. In our work, we build upon the win-rate and lift the above requirement by using pairwise comparisons made by a strong LLM.

## 2 LLM Ranking under Uncertainty

Let $\mathcal{M}$ be a set of $k$ large language models (LLMs) and $P(Q)$ be a distribution of inputs on a discrete set of inputs $\mathcal{Q}$. Moreover, assume that, for each input $q \sim P(Q)$,[1] each model $m \in \mathcal{M}$ may provide an output $r \sim P_m(R \mid Q = q)$ from a discrete set of outputs $\mathcal{R}$. Further, given two outputs $r, r' \in \mathcal{R}$ from two different models, the (binary) variables $w, w' \sim P(W, W' \mid Q = q, R = r, R' = r')$ indicate whether a human prefers $r$ over $r'$ ($w = 1$, $w' = 0$) or viceversa ($w = 0$, $w' = 1$). In the case of a tie, then $w = w' = 0$. In what follows, we use $m(r)$ and $m(r')$ to denote the models that provide outputs $r$ and $r'$ respectively, and without loss of generality, we assume that the output $r$ is shown first. Then, our goal is to rank all models according to the (empirical) probability $\theta_m$ that their outputs are preferred over the outputs of any other model picked uniformly at random.

To this end, we start by writing the probability $\theta_m$ as an expectation over the distribution of inputs, outputs and pairwise preferences:

$$
\theta_m = \frac{1}{k-1} \sum_{\tilde{m} \in \mathcal{M} \setminus \{m\}} \mathbb{E}_Q \left[ \frac{1}{2} \mathbb{E}_{R \sim P_m, R' \sim P_{\tilde{m}}} \left[ \mathbb{E}_W[W \mid Q, R, R'] \right] + \frac{1}{2} \mathbb{E}_{R \sim P_{\tilde{m}}, R' \sim P_m} \left[ \mathbb{E}_{W'}[W' \mid Q, R, R'] \right] \right],
\tag{1}
$$

where note that the order of the pairs of outputs is picked at random. Next, following previous work [48, 50], we formally characterize the ranking position of each model $m \in \mathcal{M}$ in the ranking induced by the probabilities $\theta_m$ using a rank-set $[l(m), u(m)]$, where

$$
l(m) = 1 + \sum_{\tilde{m} \in \mathcal{M} \setminus \{m\}} \mathbf{1}\{\theta_m < \theta_{\tilde{m}}\} \quad \text{and} \quad u(m) = k - \sum_{\tilde{m} \in \mathcal{M} \setminus \{m\}} \mathbf{1}\{\theta_m > \theta_{\tilde{m}}\},
\tag{2}
$$

are the lower and upper ranking position respectively and smaller ranking position indicates better alignment with human preferences. Here, note that it often holds that $\theta_m \neq \theta_{\tilde{m}}$ for all $\tilde{m} \in \mathcal{M} \setminus \{m\}$ and then the rank-set reduces to a singleton, *i.e.*, $l(m) = u(m)$.

---

[1]We denote random variables with capital letters and realizations of random variables with lower case letters.

In general, we cannot directly construct the rank-sets as defined above because the probabilities $\theta_m$ are unknown. Consequently, the typical strategy reduces to first gathering pairwise comparisons by humans to compute unbiased estimates of the above probabilities using sample averages and then construct estimates $[\hat{l}(m), \hat{u}(m)]$ of the rank-sets using Eq. 2 with $\hat{\theta}_m$ rather than $\theta_m$. Under this strategy, if the amount of pairwise comparisons we gather is sufficiently large, the estimates of the rank-sets will closely match the true rank-sets. However, since gathering pairwise comparisons from humans is costly and time-consuming, it has become a very common practice to gather pairwise comparisons $\hat{w}, \hat{w}'$ by a strong LLM, rather than pairwise comparisons $w, w'$ by humans [12–14, 28, 29, 31, 61–64], and then utilize them to compute unbiased estimates of the probabilities $\breve{\theta}_m$ that the outputs provided by each model is preferred over others by the strong LLM, which can be written in terms of expectations as follows:

$$
\begin{aligned}
\breve{\theta}_m = \frac{1}{k-1} \sum_{\tilde{m} \in \mathcal{M} \setminus \{m\}} \mathbb{E}_Q \Bigg[ & \frac{1}{2} \mathbb{E}_{R \sim P_m, R' \sim P_{\tilde{m}}} \Big[ \mathbb{E}_{\hat{W}}[\hat{W} \,|\, Q, R, R'] \Big] + \\
& \frac{1}{2} \mathbb{E}_{R \sim P_{\tilde{m}}, R' \sim P_m} \Big[ \mathbb{E}_{\hat{W}'}[\hat{W}' \,|\, Q, R, R'] \Big] \Bigg].
\end{aligned}
\tag{3}
$$

In general, one can only draw valid conclusions about $\boldsymbol{\theta}$ using (an estimate of) $\breve{\boldsymbol{\theta}}$ if the distribution of the pairwise comparisons by the strong LLM $P(\hat{W}, \hat{W}' \,|\, Q = q, R = r, R' = r')$ closely matches the distribution of pairwise comparisons by the humans $P(W, W' \,|\, Q = q, R = r, R' = r')$ for any $q \in \mathcal{Q}$ and $r, r' \in \mathcal{R}$. However, there are multiple lines of evidence showing that there is a mismatch between the distributions, questioning the validity of the conclusions drawn by a myriad of papers. In what follows, we introduce a statistical framework that, by complementing a (large) set of $N + n$ pairwise comparisons $\hat{w}, \hat{w}'$ by a strong LLM with a small set of $n$ pairwise comparisons $w, w'$ by humans, is able to construct estimates $[\hat{l}(m), \hat{u}(m)]$ of the rank-sets with provable coverage guarantees. More formally, given a user-specified value $\alpha \in (0, 1)$, the estimates of the rank-sets satisfy that

$$
\lim_n \mathbb{P} \left( \bigcap_{m \in \mathcal{M}} [l(m), u(m)] \subseteq [\hat{l}(m), \hat{u}(m)] \right) \geq 1 - \alpha.
\tag{4}
$$

To this end, we will first use prediction-powered inference [37, 38] to construct a confidence ellipsoid that, with probability $1 - \alpha$, is guaranteed to contain the (column) vector of (true) probabilities $\boldsymbol{\theta} = (\theta_m)_{m \in \mathcal{M}}$. Then, we will use the distance between this ellipsoid and the hyperplanes under which each pair of models $m, \tilde{m} \in \mathcal{M}$ have the same probability values of being preferred over others, to efficiently construct the estimates $[\hat{l}(m), \hat{u}(m)]$ of the rank-sets.

## 3 Constructing Confidence Regions with Prediction-Powered Inference

Let the set $\mathcal{D}_N = \{(q_i, r_i, r_i', m(r_i), m(r_i'), \hat{w}_i, \hat{w}_i')\}_{i=1}^N$ comprise pairwise comparisons by a strong LLM to $N$ inputs and the set $\mathcal{D}_n = \{(q_i, r_i, r_i', m(r_i), m(r_i'), w_i, w_i', \hat{w}_i, \hat{w}_i')\}_{i=1}^n$ comprise pairwise comparisons by the same strong LLM and by humans to $n$ inputs, with $n \ll N$. In what follows, for each pairwise comparison, we will refer to the models $m(r)$ and $m(r')$ that provided the first and second output using one-hot (column) vectors $\boldsymbol{m}$ and $\boldsymbol{m}'$, respectively. Moreover, to summarize the pairwise comparisons[2] in $\mathcal{D}_N$ and $\mathcal{D}_n$, we will stack the one-hot vectors $\boldsymbol{m}$ and $\boldsymbol{m}'$ into four matrices, $\boldsymbol{M}_N$ and $\boldsymbol{M}_N'$ for $\mathcal{D}_N$ and $\boldsymbol{M}_n$ and $\boldsymbol{M}_n'$ for $\mathcal{D}_n$, where each column corresponds to a one-hot vector, and the indicators $w$ and $\hat{w}$ into six (column) vectors, $\hat{\boldsymbol{w}}_N$ and $\hat{\boldsymbol{w}}_N'$ for $\mathcal{D}_N$ and $\hat{\boldsymbol{w}}_n$, $\hat{\boldsymbol{w}}_n'$, $\boldsymbol{w}_n$ and $\boldsymbol{w}_n'$ for $\mathcal{D}_n$.

---

[2]We assume that each model $m \in \mathcal{M}$ participates in at least one pairwise comparison in both $\mathcal{D}_N$ and $\mathcal{D}_n$.

---

**Algorithm 1:** It estimates $\hat{\boldsymbol{\theta}}$ and $\widehat{\boldsymbol{\Sigma}}$ using prediction-powered inference.

---

**Input:** $k, \mathcal{D}_N, \mathcal{D}_n$
**Output:** $\hat{\boldsymbol{\theta}}, \widehat{\boldsymbol{\Sigma}}$

1   $\hat{\boldsymbol{w}}_N, \hat{\boldsymbol{w}}'_N, \boldsymbol{M}_N, \boldsymbol{M}'_N \leftarrow \text{SUMMARIZE}(\mathcal{D}_N, k)$

2   $\boldsymbol{w}_n, \boldsymbol{w}'_n, \hat{\boldsymbol{w}}_n, \hat{\boldsymbol{w}}'_n, \boldsymbol{M}_n, \boldsymbol{M}'_n \leftarrow \text{SUMMARIZE}(\mathcal{D}_n, k)$

3   $\lambda \leftarrow \text{LAMBDA}(k, \hat{\boldsymbol{w}}_N, \hat{\boldsymbol{w}}'_N, \boldsymbol{M}_N, \boldsymbol{M}'_N, \boldsymbol{w}_n, \boldsymbol{w}'_n, \hat{\boldsymbol{w}}_n, \hat{\boldsymbol{w}}'_n, \boldsymbol{M}_n, \boldsymbol{M}'_n)$       `// Algorithm 4`

4   $\boldsymbol{a} \leftarrow \left(\mathbf{1}_k\left((\boldsymbol{M}_N + \boldsymbol{M}'_N)\,\mathbf{1}_N\right)^\top \odot \mathbb{I}_k\right)^{-1}(\boldsymbol{M}_N \cdot \lambda\hat{\boldsymbol{w}}_N + \boldsymbol{M}'_N \cdot \lambda\hat{\boldsymbol{w}}'_N)$

5   $\boldsymbol{b} \leftarrow \left(\mathbf{1}_k\left((\boldsymbol{M}_n + \boldsymbol{M}'_n)\,\mathbf{1}_n\right)^\top \odot \mathbb{I}_k\right)^{-1}(\boldsymbol{M}_n(\lambda\hat{\boldsymbol{w}}_n - \boldsymbol{w}_n) + \boldsymbol{M}'_n(\lambda\hat{\boldsymbol{w}}'_n - \boldsymbol{w}'_n))$

6   $\hat{\boldsymbol{\theta}} \leftarrow \boldsymbol{a} - \boldsymbol{b}$       `// prediction-powered estimate (Eq. 5)`

7   $\boldsymbol{A} \leftarrow \left(\left(\mathbf{1}_k(\lambda\hat{\boldsymbol{w}}_N - \boldsymbol{M}_N^\top\boldsymbol{a})^\top\right) \odot \boldsymbol{M}_N + \left(\mathbf{1}_k(\lambda\hat{\boldsymbol{w}}'_N - \boldsymbol{M}_N'^\top\boldsymbol{a})^\top\right) \odot \boldsymbol{M}'_N\right)$

8   $\boldsymbol{B} \leftarrow \left(\mathbf{1}_k(\lambda\hat{\boldsymbol{w}}_n - \boldsymbol{w}_n - \boldsymbol{M}_n^\top\boldsymbol{b})^\top\right) \odot \boldsymbol{M}_n + \left(\mathbf{1}_k(\lambda\hat{\boldsymbol{w}}'_n - \boldsymbol{w}'_n - \boldsymbol{M}_n'^\top\boldsymbol{b})^\top\right) \odot \boldsymbol{M}'_n$

9   $\widehat{\boldsymbol{\Sigma}} \leftarrow \frac{1}{N^2}\boldsymbol{A}\boldsymbol{A}^\top + \frac{1}{n^2}\boldsymbol{B}\boldsymbol{B}^\top$       `// estimate of covariance (Eq. 7)`

10   **return** $\hat{\boldsymbol{\theta}}, \widehat{\boldsymbol{\Sigma}}$

---

Then, building upon the recent framework of prediction-powered inference [37], we compute an unbiased estimate $\hat{\boldsymbol{\theta}}$ of the vector of (true) probabilities $\boldsymbol{\theta}$:

$$\hat{\boldsymbol{\theta}} = \underbrace{\left(\mathbf{1}_k\left((\boldsymbol{M}_N + \boldsymbol{M}'_N)\,\mathbf{1}_N\right)^\top \odot \mathbb{I}_k\right)^{-1}(\boldsymbol{M}_N \cdot \lambda\hat{\boldsymbol{w}}_N + \boldsymbol{M}'_N \cdot \lambda\hat{\boldsymbol{w}}'_N)}_{\boldsymbol{a}}$$

$$- \underbrace{\left(\mathbf{1}_k\left((\boldsymbol{M}_n + \boldsymbol{M}'_n)\,\mathbf{1}_n\right)^\top \odot \mathbb{I}_k\right)^{-1}(\boldsymbol{M}_n(\lambda\hat{\boldsymbol{w}}_n - \boldsymbol{w}_n) + \boldsymbol{M}'_n(\lambda\hat{\boldsymbol{w}}'_n - \boldsymbol{w}'_n))}_{\boldsymbol{b}}, \quad (5)$$

where $\mathbf{1}_d$ denotes a $d$-dimensional column vector where each dimension has value 1 and $\mathbb{I}_k$ denotes a $k$-dimensional identity matrix. Here, note that the first term $\boldsymbol{a}$ utilizes the pairwise comparisons by the strong LLM from $\mathcal{D}_N$ to compute an unbiased estimate of the vector of probabilities $\breve{\boldsymbol{\theta}}$ defined in Eq. 3 using sample averages, and the second term $\boldsymbol{b}$ utilizes the pairwise comparisons by the strong LLM and by humans from $\mathcal{D}_n$ to compute an unbiased estimate of the difference of probabilities $\boldsymbol{\theta} - \breve{\boldsymbol{\theta}}$ defined in Eqs. 1 and 3, also using sample averages. The parameter $\lambda \in [0, 1]$ weighs the comparisons $\hat{\boldsymbol{w}}, \hat{\boldsymbol{w}}'$ differently than the comparisons $\boldsymbol{w}, \boldsymbol{w}'$. Details on why this can be useful and on the selection of $\lambda$ are in Appendix B.2.

Further, as shown in Angelopoulos et al. [38], the difference of probabilities $\hat{\boldsymbol{\theta}} - \boldsymbol{\theta}$ converges in distribution to a $k$-dimensional normal $\mathcal{N}_k(0, \boldsymbol{\Sigma})$, where $\boldsymbol{\Sigma} = \mathbb{E}[(\hat{\boldsymbol{\theta}} - \boldsymbol{\theta})(\hat{\boldsymbol{\theta}} - \boldsymbol{\theta})^\top]$, and thus the confidence region

$$\mathcal{C}_\alpha = \left\{\boldsymbol{x} \in \mathbb{R}^k \mid \left(\boldsymbol{x} - \hat{\boldsymbol{\theta}}\right)^\top \left(\frac{\widehat{\boldsymbol{\Sigma}}^{-1}}{\chi^2_{k,1-\alpha}}\right) \left(\boldsymbol{x} - \hat{\boldsymbol{\theta}}\right) \leq 1\right\}, \quad (6)$$

where $\widehat{\boldsymbol{\Sigma}}$ is an empirical estimate of the covariance matrix $\boldsymbol{\Sigma}$ using pairwise comparisons from $\mathcal{D}_N$ and $\mathcal{D}_n$, i.e.,

$$\widehat{\boldsymbol{\Sigma}} = \frac{1}{N^2}\boldsymbol{A}\boldsymbol{A}^\top + \frac{1}{n^2}\boldsymbol{B}\boldsymbol{B}^\top, \quad (7)$$

with

$$\boldsymbol{A} = \left(\left(\mathbf{1}_k(\lambda\hat{\boldsymbol{w}}_N - \boldsymbol{M}_N^\top\boldsymbol{a})^\top\right) \odot \boldsymbol{M}_N + \left(\mathbf{1}_k(\lambda\hat{\boldsymbol{w}}'_N - \boldsymbol{M}_N'^\top\boldsymbol{a})^\top\right) \odot \boldsymbol{M}'_N\right),$$

$$\boldsymbol{B} = \left(\mathbf{1}_k(\lambda\hat{\boldsymbol{w}}_n - \boldsymbol{w}_n - \boldsymbol{M}_n^\top\boldsymbol{b})^\top\right) \odot \boldsymbol{M}_n + \left(\mathbf{1}_k(\lambda\hat{\boldsymbol{w}}'_n - \boldsymbol{w}'_n - \boldsymbol{M}_n'^\top\boldsymbol{b})^\top\right) \odot \boldsymbol{M}'_n,$$

and $\chi^2_{k,1-\alpha}$ is the $1 - \alpha$ quantile of the $\chi^2$ distribution with $k$ degrees of freedom, satisfies that

$$\lim_n \mathbb{P}(\boldsymbol{\theta} \in \mathcal{C}_\alpha) = 1 - \alpha. \quad (8)$$

Algorithm 1 summarizes the overall procedure to compute $\hat{\boldsymbol{\theta}}$ and $\widehat{\boldsymbol{\Sigma}}$, which runs in $O(k^2(N + n))$ time.

**Algorithm 2:** It constructs $[\hat{l}(m), \hat{u}(m)]$ for all $m \in \mathcal{M}$

---

**Input:** $\mathcal{M}, \mathcal{D}_N, \mathcal{D}_n, \alpha$
**Output:** $\{[\hat{l}(m), \hat{u}(m)]\}_{m \in \mathcal{M}}$

**1** $k \leftarrow |\mathcal{M}|$
**2** $\hat{\boldsymbol{\theta}}, \widehat{\boldsymbol{\Sigma}} \leftarrow \text{CONFIDENCE-ELLIPSOID}(k, \mathcal{D}_N, \mathcal{D}_n)$         `// Algorithm 1`
**3** **for** $m \in \mathcal{M}$ **do**
**4**      $\hat{l}(m) \leftarrow 1, \quad \hat{u}(m) \leftarrow k$
**5**      **for** $\tilde{m} \in \mathcal{M} \setminus \{m\}$ **do**
**6**          $d \leftarrow \frac{|\hat{\theta}_m - \hat{\theta}_{\tilde{m}}|}{\sqrt{2}} - \sqrt{\frac{1}{2}(\widehat{\Sigma}_{m,m} + \widehat{\Sigma}_{\tilde{m},\tilde{m}} - 2\widehat{\Sigma}_{m,\tilde{m}})\chi^2_{k,1-\alpha}}$     `// Eq. 10`
**7**          **if** $d > 0$ **and** $\hat{\theta}_m < \hat{\theta}_{\tilde{m}}$ **then**
**8**             $\hat{l}(m) \leftarrow \hat{l}(m) + 1$
**9**          **else if** $d > 0$ **and** $\hat{\theta}_m > \hat{\theta}_{\tilde{m}}$ **then**
**10**             $\hat{u}(m) \leftarrow \hat{u}(m) - 1$

**11** **return** $\{[\hat{l}(m), \hat{u}(m)]\}_{m \in \mathcal{M}}$

---

## 4 Constructing Rank-Sets with Coverage Guarantees

For each pair of models $m, \tilde{m} \in \mathcal{M}$ such that $m \neq \tilde{m}$, we first define a hyperplane $H_{m,\tilde{m}} \subseteq \mathbb{R}^k$ as follows:

$$H_{m,\tilde{m}} = \{\boldsymbol{x} \in \mathbb{R}^k \,|\, x_m = x_{\tilde{m}}\}. \tag{9}$$

Then, for each of these hyperplanes $H_{m,\tilde{m}}$, we calculate the distance $d(\mathcal{C}_\alpha, H_{m,\tilde{m}})$ between $H_{m,\tilde{m}}$ and the confidence region $\mathcal{C}_\alpha$ defined by Eq. 6, *i.e.*,

$$d(\mathcal{C}_\alpha, H_{m,\tilde{m}}) = \frac{|\hat{\theta}_m - \hat{\theta}_{\tilde{m}}| - \sqrt{(\widehat{\Sigma}_{m,m} + \widehat{\Sigma}_{\tilde{m},\tilde{m}} - 2\widehat{\Sigma}_{m,\tilde{m}})\chi^2_{k,1-\alpha}}}{\sqrt{2}}, \tag{10}$$

where $\widehat{\boldsymbol{\Sigma}}$ is the empirical covariance matrix defined by Eq. 7.

Now, for each pair of models $m, m' \in \mathcal{M}$, we can readily conclude that, if the distance $d(\mathcal{C}_\alpha, H_{m,\tilde{m}}) > 0$, then, the confidence region $\mathcal{C}_\alpha$ either lies in the half-space of $\mathbb{R}^k$ where $x_m > x_{\tilde{m}}$ if $\hat{\theta}_m > \hat{\theta}_{\tilde{m}}$ or it lies in the half space of $\mathbb{R}^k$ where $x_m < x_{\tilde{m}}$ if $\hat{\theta}_m < \hat{\theta}_{\tilde{m}}$. Building upon this observation, for each model $m \in \mathcal{M}$, we construct the following estimates $[\hat{l}(m), \hat{u}(m)]$ of the rank-sets $[l(m), u(m)]$:

$$
\begin{aligned}
\hat{l}(m) &= 1 + \sum_{\tilde{m} \in \mathcal{M} \setminus \{m\}} \mathbf{1}\{d(\mathcal{C}_\alpha, H_{m,\tilde{m}}) > 0\} \cdot \mathbf{1}\{\hat{\theta}_m < \hat{\theta}_{\tilde{m}}\} \\
\hat{u}(m) &= k - \sum_{\tilde{m} \in \mathcal{M} \setminus \{m\}} \mathbf{1}\{d(\mathcal{C}_\alpha, H_{m,\tilde{m}}) > 0\} \cdot \mathbf{1}\{\hat{\theta}_m > \hat{\theta}_{\tilde{m}}\}.
\end{aligned}
\tag{11}
$$

Importantly, using a similar proof technique as in Lemma 1 in Neuhof and Benjamini [48], we can show that the above rank-sets estimates enjoy provable coverage guarantees with respect to the rank-sets $[l(m), u(m)]$ induced by the probabilities $\boldsymbol{\theta}$ that the outputs of each model is preferred over any other model by humans (proven in Appendix A):

**Theorem 4.1** *The estimates $[\hat{l}(m), \hat{u}(m)]$ of the rank-sets defined by Eq. 11 satisfy that*

$$\lim_n \mathbb{P}\left( \bigcap_{m \in \mathcal{M}} [l(m), u(m)] \subseteq [\hat{l}(m), \hat{u}(m)] \right) \geq 1 - \alpha. \tag{12}$$

Algorithm 2 summarizes the overall procedure to construct the rank-sets $[\hat{l}(m), \hat{u}(m)]$ for all $m \in \mathcal{M}$, which runs in $O(k^2(N + n))$.

# 5 Experiments

We apply our framework to construct rank-sets for 12 popular LLMs using pairwise comparisons made by humans in the LMSYS Chatbot Arena platform[3] and pairwise comparisons made by three strong LLMs. We show that the rank-sets constructed using our framework are significantly more likely to cover the true ranking consistent with (the distribution of) human pairwise preferences than the rank-sets constructed using only pairwise comparisons made by the strong LLMs.

**Experimental setup.** Our starting point is the Chatbot Arena dataset [12], which comprises $33,481$ pairwise comparisons made by $13,383$ humans about the responses given by 20 different LLMs to $26,968$ unique queries. In what follows, we refer to each pair of responses to a query by two different LLMs and the query itself as an instance. As an initial pre-processing, we filter out any instance whose corresponding query is flagged as toxic or multiturn. Then, we gather pairwise comparisons made by three strong LLMs, namely GPT-3.5-turbo-0125 (GPT3.5), GPT-4-0125-preview (GPT4) and Claude-3-Opus-20240229 (CL3), about all the (pre-processed) instances from the Chatbot Arena dataset. To this end, we use (almost) the same prompt as in Zheng et al. [12], which instructs each strong LLM to output option 'A' ('B') if it prefers the response of first (second) LLM, or option 'C' if it declares a tie. Further, we filter out any instances for which at least one strong LLM provides a verbose output instead of 'A', 'B', or 'C', and focus on a set of LLMs with at least 96 pairwise comparisons between every pair of LLMs in the set. After these pre-processing steps, we have $14,947$ instances comprising $13,697$ unique queries and 12 different LLMs and, for each instance, we have one pairwise comparison made by a human and three pairwise comparisons by the three strong LLMs. Refer to Appendix C for more information regarding the 12 LLMs, the number of pairwise comparisons between every pair of LLMs, and the prompt used to gather pairwise comparisons made by the three strong LLMs.

To draw reliable conclusions, in each experiment, we construct rank-sets $1,000$ times and, each time, we use a random set of $N + n = 6,336$ instances with an equal number of instances per pair of models, out of the $14,947$ instances. The values of $N$ and $n$ vary across experiments and they define two random subsets, also with an equal number of instances per pair of models.

**Methods.** In our experiments, we construct rank-sets using the following methods:

a) BASELINE: it constructs (unbiased) rank-sets using the pairwise comparisons made by humans corresponding to the random set of $N + n$ instances via Algorithms 2 and 3, shown in Appendix B.1. The constructed rank-sets are presumably likely to cover the true rank-sets.

b) LLM GPT4, LLM GPT3.5 and LLM CL3: they construct (possibly biased) rank-sets using the pairwise comparisons made by one of the three strong LLMs corresponding to the random set of $N + n$ instances via Algorithms 2 and 3.

c) PPR GPT4, PPR GPT3.5 and PPR CL3: they construct (unbiased) rank-sets using pairwise comparisons made by one of the three strong LLMs corresponding to the random set of $N + n$ instances and pairwise comparisons made by humans corresponding to the random subset of $n$ instances via Algorithms 1 and 2.

d) HUMAN ONLY: it constructs (unbiased) rank-sets using the pairwise comparisons made by humans corresponding to the random subset of $n$ instances via Algorithms 2 and 3.

In the above, note that a), b) and d) use linear regression to construct a confidence region $\mathcal{C}_\alpha$ using only pairwise comparisons by either humans or a strong LLM via Algorithm 3, which runs in $O(k^2(N + n))$ in a)-b) and in $O(k^2n)$ in d), and then use this confidence region to construct the rank-sets via Algorithm 2.

**Quality metrics.** Since the true probabilities $\boldsymbol{\theta}$ are unknown, we cannot compute the true rank-sets of the 12 LLMs under comparison, which presumably may be singletons. As a result, we cannot estimate the (empirical) coverage probability—the probability that the rank-sets constructed using the above methods cover the true rank-sets, which Theorem 4.1 refers to. To overcome this, we assess the quality of the rank-sets using two alternative metrics: rank-set size and baseline intersection probability. Here, smaller (larger) rank-set sizes and larger (smaller) intersection probabilities are better (worse). The baseline intersection probability is just the (empirical) probability that the rank-sets $[\hat{l}(m), \hat{u}(m)]$

---

[3]https://chat.lmsys.org/

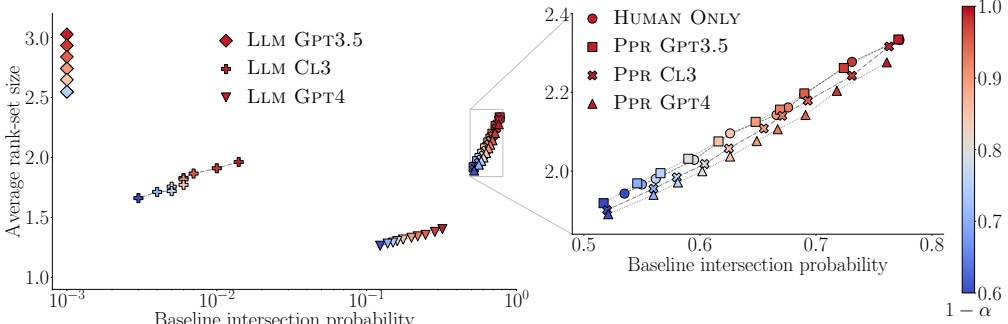

Figure 1: Average rank-set size against baseline intersection probability for rank-sets constructed using only pairwise comparisons by a strong LLM (LLM GPT4, LLM GPT3.5 and LLM CL3), only pairwise comparisons by humans (HUMAN ONLY), and pairwise comparisons by both a strong LLM and humans (PPR GPT4, PPR GPT3.5 and PPR CL3) for different values of $\alpha$ and $n = 990$. Smaller (larger) average rank-set sizes and larger (smaller) intersection probabilities are better (worse). In all panels, 95% confidence bars for the rank-set size are not shown, as they are always below 0.02.

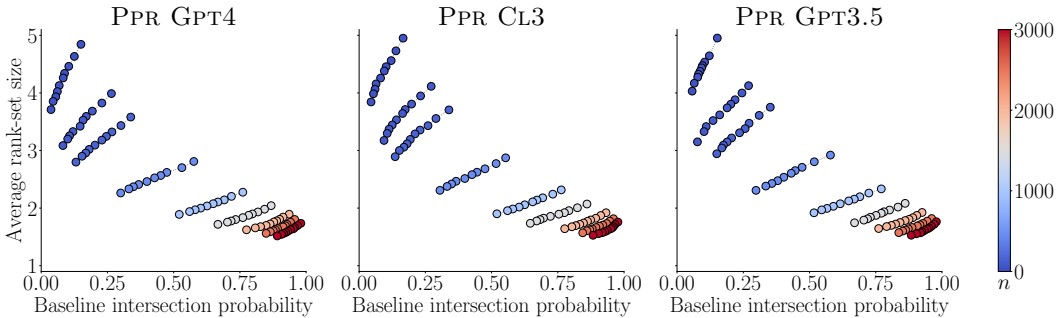

Figure 2: Average rank-set size against baseline intersection probability for rank-sets constructed using pairwise comparisons by both a strong LLM and humans for different values of $n$ and $\alpha$. Smaller (larger) average rank-set sizes and larger (smaller) intersection probabilities are better (worse). In all panels, 95% confidence bars for the rank-set size are not shown, as they are always below 0.04.

constructed using one of the above methods intersect with the rank-sets $[\tilde{l}(m), \tilde{u}(m)]$ constructed using the BASELINE method, *i.e.*, $\mathbb{P}\left(\bigcap_{m \in \mathcal{M}} \mathbf{1}\left\{[\tilde{l}(m), \tilde{u}(m)] \cap [\hat{l}(m), \hat{u}(m)]\right\}\right)$. Intuitively, we expect that the larger the baseline intersection probability, the larger the coverage probability since the BASELINE method uses a large(r) number of pairwise comparisons by humans to construct (unbiased) rank-sets and thus it is expected to approximate well the true rank-sets. Further, note that the baseline intersection probability tells us how frequently there exists at least one single ranking covered by both one of the above methods and the BASELINE method. In Appendix D.2, we experiment with an alternative metric, namely baseline coverage probability, which is the (empirical) probability that the rank-sets constructed using one of the above methods covers the rank-sets constructed using the BASELINE method. In Appendix E, we additionally evaluate our framework in a synthetic setting where the true rank-sets are known, allowing us to compute the coverage probability and rank-biased overlap (RBO) [65].

**Quality of the rank-sets.** Figure 1 shows the average rank-set size against the baseline intersection probability for rank-sets constructed using all methods[4] except BASELINE for different $\alpha$ values[5] and $n = 990$. The results show several interesting insights. First, we find that rank-sets constructed using only pairwise comparisons by a strong LLM (LLM GPT4, LLM GPT3.5 and LLM CL3) achieve much lower baseline intersection probability, even orders of magnitude lower, than those constructed using only pairwise comparisons by humans (HUMAN ONLY) or using both pairwise comparisons by a

---

[4]In Appendix D.1, we include a version of this figure with three panels, where each panel contains the results for one strong LLM and confidence regions for the rank-set sizes.

[5]$\alpha \in \{0.4, 0.3, 0.25, 0.2, 0.15, 0.1, 0.075, 0.05, 0.025, 0.01\}$.

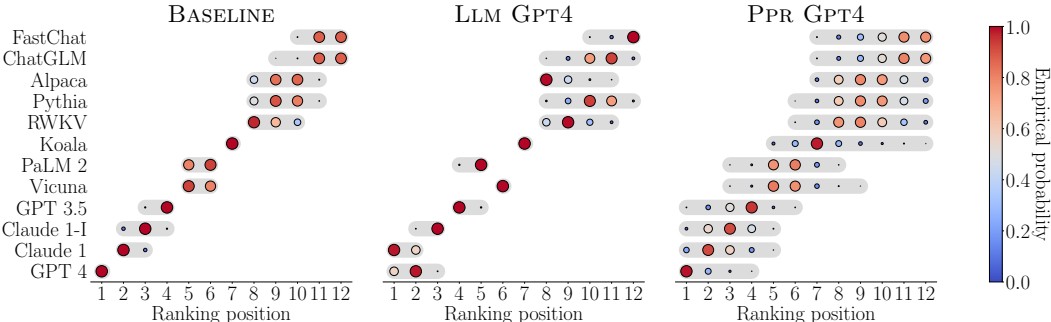

Figure 3: Empirical probability that each ranking position is included in the rank-sets constructed by BASELINE, LLM GPT4 and PPR GPT4 for each of the LLMs under comparison. In all panels, $n = 990$ and $\alpha = 0.05$. Larger (smaller) dots indicate higher (lower) empirical probability.

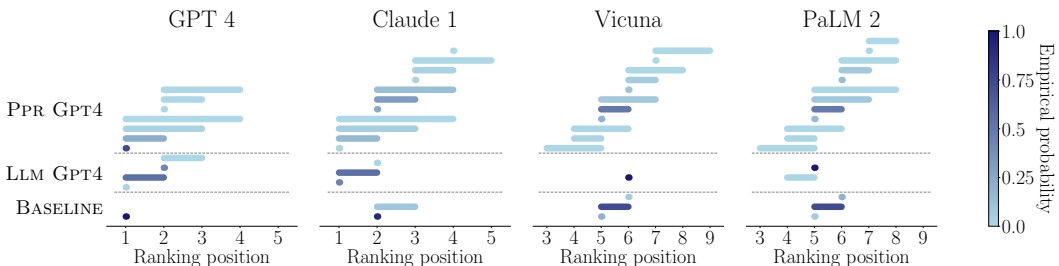

Figure 4: Empirical probability of each rank-set constructed by BASELINE, LLM GPT4 and PPR GPT4 for GPT 4 (left), Claude 1 (middle left), Vicuna (middle right) and PaLM 2 (right). In all panels, $n = 990$ and $\alpha = 0.05$.

strong LLM and humans (PPR GPT4, PPR GPT3.5 and PPR CL3). This suggests that the distributions of pairwise comparisons by strong LLMs and humans are actually different, questioning the rationale used by an extensive line of work that proposed using only pairwise comparisons by strong LLMs to rank LLMs [12, 25–29, 31]. Second, we find that rank-sets constructed using both pairwise comparisons by two of the strong LLMs and humans (PPR GPT4 and PPR CL3) achieve a better trade-off between average rank-set size and baseline intersection probability than those constructed using only pairwise comparisons by humans (HUMAN ONLY). This suggests that pairwise comparisons by strong LLMs are valuable if they are complemented with (a few) pairwise comparisons by humans. Third, we find that, among the three strong LLMs, GPT 4 stands out as the best performer.

Figure 2 shows the average rank-set size against the baseline intersection probability for rank-sets constructed using PPR GPT4, PPR GPT3.5 and PPR CL3 for different values of $n$ and $\alpha$ (the same values as in Figure 1).[6] The results show that the trade-off between rank-sets and baseline intersection probabilities improves rapidly as the number of pairwise comparisons by humans $n$ increases but with diminishing returns.

**Structure of the rank-sets.** In this section, we take a closer look to the structure of the rank-sets constructed using BASELINE, LLM GPT4 and PPR GPT4. In Appendix D.3, we include additional results for all other methods.

First, we compute the empirical probability that each ranking position is included in the rank-sets constructed by BASELINE, LLM GPT4 and PPR GPT4 of each of the LLMs under comparison. Figure 3 summarizes the results for $n = 990$ and $\alpha = 0.05$, which reveal several interesting insights. We find that there is lower uncertainty regarding the ranking position of each model for LLM GPT4 than for PPR GPT4. However, for LLM GPT4, the ranking position with the highest probability mass differs from BASELINE in 7 out of 12 LLMs, including the top-2 performers. In contrast, for PPR GPT4, it only differs from BASELINE in 3 out of 12 LLMs. This questions once more the status quo, which proposed using only pairwise comparisons by strong LLMs to rank LLMs [12, 25–29, 31].

---

[6] $n \in \{66, 132, 198, 462, 990, 1452, 1980, 2442, 2970\}$.

Next, we compute the empirical probability of each rank-set constructed by Baseline, Llm Gpt4 and Ppr Gpt4 for each of the LLMs under comparison. Figure 4 summarizes the results for GPT 4, Claude 1, Vicuna and PaLM 2 for $n = 990$ and $\alpha = 0.05$. In agreement with the findings derived from Figure 3, we observe that the distribution of rank-sets constructed by Llm Gpt4 is more concentrated than the distribution of rank sets constructed by Ppr Gpt4. However, the rank-sets with the highest probability mass constructed by Llm Gpt4 coincide with those constructed by Baseline much less frequently than those constructed by Ppr Gpt4. Refer to Appendix D.3 for qualitatively similar results for other LLMs.

## 6 Discussion and Limitations

In this section, we highlight several limitations of our work, discuss its broader impact, and propose avenues for future work.

**Data.** Our framework assumes that the queries and the pairwise comparisons made by humans and the strong LLMs are drawn i.i.d. from fixed distributions. In future work, it would be very interesting to lift these assumptions and allow for distribution shift. Moreover, our framework assumes that the pairwise comparisons made by humans are truthful. However, an adversary could have an economic incentive to make pairwise comparisons strategically in order to favor a specific model over others. In this context, it would be interesting to extend our framework so that it is robust to strategic behavior.

**Methodology.** Our framework utilizes rank-sets as a measure of uncertainty in rankings. However, in case of limited pairwise comparison data, rank-sets may be large and overlapping, reducing their value. In such situations, it may be worthwhile to explore other measures of uncertainty for rankings beyond rank-sets. Further, to measure the level of alignment with human preferences, our framework utilizes the win-rate—the probability that the outputs of each model are preferred over the outputs of any other model picked uniformly at random. However, if we need to rank $k$ LLMs and $k$ is *large*, win-rate may be impractical since, to obtain reliable estimates, we need to gather $O(k^2)$ pairwise comparisons made by humans. Finally, our framework constructs rank-sets with asymptotic coverage guarantees, however, it would be interesting to derive PAC-style, finite-sample coverage guarantees.

**Evaluation.** We have showcased our framework using pairwise comparisons made by humans in a single platform, namely LMSYS Chatbot Arena, and pairwise comparisons made by just three strong LLMs. As a result, one may question the generalizability of the conclusion derived from the rank-sets estimated using our framework. In this context, it is also important to acknowledge that, in LMSYS Chatbot Arena, the queries are chosen by the humans who make pairwise comparisons and this may introduce a variety of biases. Therefore, it would be interesting to apply our framework to human data from other platforms.

**Broader Impact.** Our framework rank LLMs according to their level of alignment with human preferences—a LLM is ranked higher than others if its outputs are more frequently preferred by humans. However, in many application domains, especially in high-stakes scenarios, it may be important to account for other important factors such as accuracy, fairness, bias and toxicity [57].

## 7 Conclusions

We have introduced a statistical framework to construct a ranking of a collection of LLMs consistent with their level of alignment with human preferences using a small set of pairwise comparisons by humans and a large set of pairwise comparisons by a strong LLM. Our framework quantifies uncertainty in the ranking by providing a rank-set—a set of possible ranking positions—for each of the models under comparison. Moreover, it guarantees that, with a probability greater than or equal to a user-specific value, the rank-sets cover the ranking consistent with the (true) probability that each model is preferred over others by humans asymptotically. Finally, we have empirically demonstrated that the rank-sets constructed using our framework are more likely to cover the true ranking consistent with (the distribution of) human pairwise preferences than the rank-sets constructed using only pairwise comparisons made by the strong LLMs.

## Acknowledgments and Disclosure of Funding

Gomez-Rodriguez acknowledges support from the European Research Council (ERC) under the European Union's Horizon 2020 research and innovation programme (grant agreement No. 945719).

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

# A    Proof of Theorem 4.1

Note that Eq. 12 holds if and only if:

$$\lim_{n} \mathbb{P}\left(\exists m \in \mathcal{M} : [l(m), u(m)] \nsubseteq [\hat{l}(m), \hat{u}(m)]\right) \leq \alpha \tag{13}$$

Therefore, to prove the theorem, it is sufficient to prove that Eq. 13 holds. Now, to prove that Eq. 13, we first show that the probability on the left hand side of the above equation is smaller than or equal to the probability $\mathbb{P}(\boldsymbol{\theta} \notin \mathcal{C}_\alpha)$.

To this end, first note that, if for at least one model $m \in \mathcal{M}$, we have that $\hat{l}(m) > l(m)$ or $\hat{u}(m) < u(m)$, then it holds that

$$\bigcap_{m \in \mathcal{M}} [l(m), u(m)] \nsubseteq [\hat{l}(m), \hat{u}(m)].$$

Next, without loss of generality, assume that, for model $m$, we have that $\hat{l}(m) > l(m)$. In this case, from Eqs. 11 and 2 we get:

$$\sum_{\tilde{m} \in \mathcal{M} \setminus \{m\}} \mathbf{1}\{d(\mathcal{C}_\alpha, H_{m,\tilde{m}}) > 0\} \cdot \mathbf{1}\{\hat{\theta}_m < \hat{\theta}_{\tilde{m}}\} \quad > \quad \sum_{\tilde{m} \in \mathcal{M} \setminus \{m\}} \mathbf{1}\{\theta_m < \theta_{\tilde{m}}\},$$

which means that there must be at least one model $\tilde{m} \in \mathcal{M}$ such that $x_m < x_{\tilde{m}} \ \forall \boldsymbol{x} \in \mathcal{C}_\alpha$ and $\theta_m > \theta_{\tilde{m}}$, which implies that $\boldsymbol{\theta} \notin \mathcal{C}_\alpha$. As a result, we can immediately conclude that,

$$\lim_{n} \mathbb{P}\left(\exists m \in \mathcal{M} : [l(m), u(m)] \nsubseteq [\hat{l}(m), \hat{u}(m)]\right) \leq \lim_{n} \mathbb{P}(\boldsymbol{\theta} \notin \mathcal{C}_\alpha) = \alpha.$$

This concludes the proof. ∎

# B Algorithms

## B.1 Algorithm 3

In this section, we present a pseudocode implementation of the algorithm to construct confidence regions using linear regression.

---

**Algorithm 3:** It estimates $\hat{\boldsymbol{\theta}}$ and $\widehat{\boldsymbol{\Sigma}}$ using linear regression.

**Input:** $k, \mathcal{D}$
**Output:** $\hat{\boldsymbol{\theta}}, \widehat{\boldsymbol{\Sigma}}$

1 $\bar{\boldsymbol{w}}, \bar{\boldsymbol{w}}', \boldsymbol{M}, \boldsymbol{M}' \leftarrow \text{SUMMARIZE}(\mathcal{D}, k)$
2 $\hat{\boldsymbol{\theta}} \leftarrow \left(\mathbf{1}_k\left((\boldsymbol{M}+\boldsymbol{M}')\,\mathbf{1}_{|\mathcal{D}|}\right)^{\top} \odot \mathbb{I}_k\right)^{-1}\left(\boldsymbol{M}\cdot\bar{\boldsymbol{w}}+\boldsymbol{M}'\cdot\bar{\boldsymbol{w}}'\right)$  `// linear reg. estimate`
3 $\boldsymbol{A} \leftarrow \left(\left(\mathbf{1}_k(\bar{\boldsymbol{w}}-\boldsymbol{M}^{\top}\hat{\boldsymbol{\theta}})^{\top}\right) \odot \boldsymbol{M} + \left(\mathbf{1}_k(\bar{\boldsymbol{w}}'-\boldsymbol{M}'^{\top}\hat{\boldsymbol{\theta}})^{\top}\right) \odot \boldsymbol{M}'\right)$
4 $\widehat{\boldsymbol{\Sigma}} \leftarrow \frac{1}{|\mathcal{D}|^2}\boldsymbol{A}\boldsymbol{A}^{\top}$  `// estimate of covariance`
5 **return** $\hat{\boldsymbol{\theta}}, \widehat{\boldsymbol{\Sigma}}$

---

Note that, in Algorithm 2, $\hat{\boldsymbol{\theta}}$ and $\widehat{\boldsymbol{\Sigma}}$ can alternatively be computed by calling Algorithm 3 instead of Algorithm 1. Algorithm 3 runs in $O(k^2|\mathcal{D}|)$ time.

## B.2 Algorithm to set $\lambda$

The parameter $\lambda \in [0,1]$ weighs the comparisons $\hat{\boldsymbol{w}}, \hat{\boldsymbol{w}}'$ by the strong LLM differently than the comparisons $\boldsymbol{w}, \boldsymbol{w}'$ by humans. This way, if the strong LLM's preferences are strongly aligned with human preferences, the pairwise comparisons by the strong LLM can be weighed close to or equally with the pairwise comparisons by humans. Conversely, if the strong LLM's preferences are not well aligned with human preferences, the pairwise comparisons $\hat{\boldsymbol{w}}, \hat{\boldsymbol{w}}'$ can be weighed lower than the pairwise comparisons $\boldsymbol{w}, \boldsymbol{w}'$. Following Angelopoulous et al. [38], we select the $\lambda$ that minimizes $\text{tr}(\boldsymbol{\Sigma})$, where $\boldsymbol{\Sigma} = \mathbb{E}[(\hat{\boldsymbol{\theta}}-\boldsymbol{\theta})(\hat{\boldsymbol{\theta}}-\boldsymbol{\theta})^{\top}]$:

$$\lambda = \frac{n}{n+N}\frac{\text{tr}(\Sigma_n)}{\text{tr}(\Sigma_N)} \tag{14}$$

where $\Sigma_N$ is the sample covariance matrix of the estimate of $\breve{\boldsymbol{\theta}}$ computed from Algorithm 3 using the pairwise comparisons by the strong LLM $\mathcal{D}_N$, and $\Sigma_n$ is the sample covariance matrix of the estimates of $\breve{\boldsymbol{\theta}}$ and $\boldsymbol{\theta}$ computed via Algorithm 3 using the pairwise comparisons by the strong LLM and by humans respectively in dataset $\mathcal{D}_n$. Detailed computation of $\lambda$ is shown in Algorithm 4, which runs in $O(k^2(N+n))$.

---

**Algorithm 4:** It computes $\lambda$

**Input:** $k, \mathcal{D}_N, \mathcal{D}_n$
**Output:** $\lambda$

1 $\boldsymbol{w}_n, \boldsymbol{w}'_n, \hat{\boldsymbol{w}}_n, \hat{\boldsymbol{w}}'_n, \boldsymbol{M}_n, \boldsymbol{M}'_n \leftarrow \text{SUMMARIZE}(\mathcal{D}_n, k)$
2 $\hat{\boldsymbol{a}}_n \leftarrow \left(\mathbf{1}_k\left((\boldsymbol{M}_n+\boldsymbol{M}'_n)\,\mathbf{1}_n\right)^{\top} \odot \mathbb{I}_k\right)^{-1}\left(\boldsymbol{M}_n\cdot\hat{\boldsymbol{w}}_n+\boldsymbol{M}'_n\cdot\hat{\boldsymbol{w}}'_n\right)$
3 $\boldsymbol{a}_n \leftarrow \left(\mathbf{1}_k\left((\boldsymbol{M}_n+\boldsymbol{M}'_n)\,\mathbf{1}_n\right)^{\top} \odot \mathbb{I}_k\right)^{-1}\left(\boldsymbol{M}_n\cdot\boldsymbol{w}_n+\boldsymbol{M}'_n\cdot\boldsymbol{w}'_n\right)$
4 $\hat{\boldsymbol{A}}_n \leftarrow \left(\left(\mathbf{1}_k(\hat{\boldsymbol{w}}_n-\boldsymbol{M}_n^{\top}\hat{\boldsymbol{a}}_n)^{\top}\right) \odot \boldsymbol{M}_n + \left(\mathbf{1}_k(\hat{\boldsymbol{w}}'_n-\boldsymbol{M}_n'^{\top}\hat{\boldsymbol{a}}_n)^{\top}\right) \odot \boldsymbol{M}'_n\right)$
5 $\boldsymbol{A}_n \leftarrow \left(\left(\mathbf{1}_k(\boldsymbol{w}_n-\boldsymbol{M}_n^{\top}\boldsymbol{a}_n)^{\top}\right) \odot \boldsymbol{M}_n + \left(\mathbf{1}_k(\boldsymbol{w}'_n-\boldsymbol{M}_n'^{\top}\boldsymbol{a}_n)^{\top}\right) \odot \boldsymbol{M}'_n\right)$
6 $\Sigma_n \leftarrow \frac{1}{n^2}\hat{\boldsymbol{A}}_n\boldsymbol{A}_n^{-1}$
7 $-, \Sigma_N \leftarrow \text{CONFIDENCE-ELLIPSOID}(k, \mathcal{D}_N)$  `// Algorithm 3`
8 $\lambda \leftarrow \frac{n}{n+N}\frac{\text{tr}(\Sigma_n)}{\text{tr}(\Sigma_N)}$
9 **return** $\lambda$

---

```
[System]
Act as an impartial judge and evaluate the responses of two AI
assistants to the user's question displayed below. Your evaluation
should consider factors such as relevance, helpfulness, accuracy,
creativity and level of detail of their responses. Output your final
verdict by strictly following this format: 'A' if the response of
assistant A is better, 'B' if the response of assistant B is better,
and 'C' for a tie. Do not give any justification or explanation for
your response.

[User]
Question:
{Query}

Response of assistant A:
{Response A}

Response of assistant B:
{Response B}
```

Listing 1: The prompt used for gathering pairwise preferences of strong LLMs.

## C  Additional Details of the Experimental Setup

In this section, we provide the implementation details and computational resources used to execute the experiments discussed in Section 5. Our algorithms are implemented in `Python 3.11.2` programming language using `NumPy` and `SciPy` open-source libraries for efficient matrix operations. Further, we use the `matplotlib` package to facilitate visualizations of our results. The complete details of the software requirements can be found in the source code provided as part of the supplementary materials. Our experiments are executed on a compute server equipped with $2 \times$ AMD EPYC 7702 processor with $64$ cores per processor and $2$ TB of main memory. It is worth noting that, our experiments are not resource intensive and can be executed on a standard desktop computer or a laptop.

**Pairwise comparisons and preprocessing.** Using the dataset from LMSYS Chatbot Arena[7], we gathered pairwise comparisons of three strong LLMs via API calls to `OpenAI API version 2024-02-15-preview` for GPT 3.5 and GPT 4, and `Anthropic API version 2023-06-01` for Claude 3. To this end, we use (almost) the same prompt as in Zheng et al. [12] that instructs each strong LLM to output option 'A' ('B') if it prefers the response of first (second) model, or option 'C' if it declares a tie. The prompt used to obtain pairwise preferences of is available in Listing 1. We preprocess the dataset by filtering out instances—an instance is a pair of responses to a query by two different models and the query itself—for which at least one strong LLM returned a verbose output instead of 'A', 'B' or 'C', and choose a set of $12$ popular large language models for our experiments. The chosen models, along with their specific versions, are listed in Table 1. In Figure 5, we show the number of pairwise comparisons per each pair of these chosen models after completing all preprocessing steps.

---

[7]The user prompts are licensed under CC-BY-4.0, while the model outputs are licensed under CC-BY-NC-4.0.

Table 1: The names and versions of the 12 popular large language models considered for our experiments after preprocessing the Chatbot Arena dataset.

| Large Language Model | Version |
|---|---|
| GPT 4 | GPT 4 |
| Claude 1 | Claude v1 |
| Claude 1-I | Claude Instant v1 |
| GPT 3.5 | GPT 3.5 turbo |
| Vicuna | Vicuna 13B |
| PaLM 2 | PaLM 2 |
| Koala | Koala 13B |
| RWKV | RWKV 4 Raven 14B |
| Pythia | OpenAssistant Pythia 12B |
| Alpaca | Alpaca 13B |
| ChatGLM | ChatGLM 6B |
| FastChat | FastChat T5 3B |

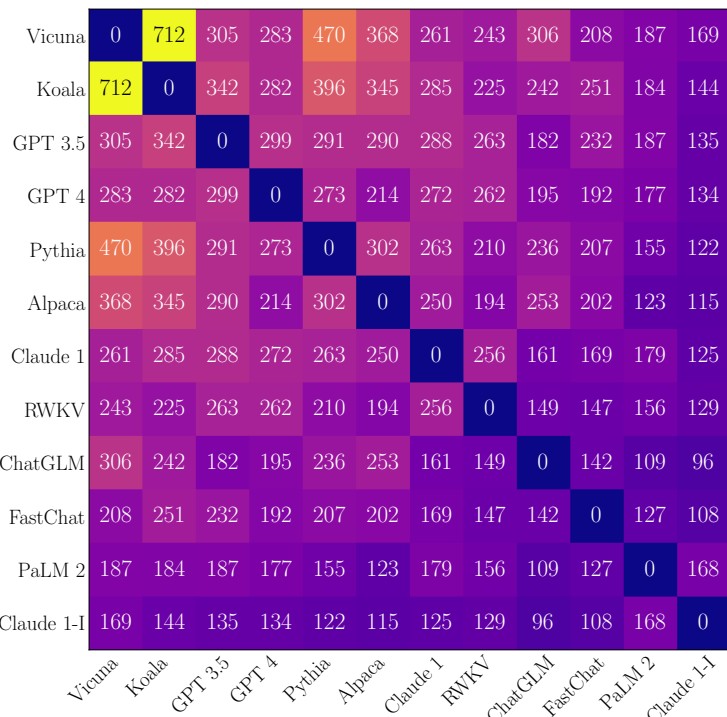

Figure 5: The number of pairwise comparisons per each pair of models after all preprocessing steps.

# D  Additional Experimental Results

In this section, we discuss additional experimental results that were omitted from the main paper due to space limitations.

## D.1  Quality of the Rank-sets

In Figure 6, we show a more detailed analysis of the average rank-set size plotted against the baseline intersection probability, with individual plots for each strong LLM: GPT4 (top), CL3 (middle) and GPT3.5 (bottom), for $n = 990$ and different $\alpha$ values.[8] All observations made in Figure 1 apply to Figure 6 as well. Moreover, it is evident that among the three strong LLMs, GPT 4 clearly outperforms the others, achieving higher baseline intersection probability and returning rank-sets with smaller average size. Specifically, in Figure 6 (a), the difference between average rank-set size between HUMAN ONLY and PPR GPT4 is significantly pronounced, while the gap gradually narrows in subsequent plots for PPR CL3 and PPR GPT3.5.

## D.2  Baseline Coverage Probability

In this subsection, we investigate a more conservative quality metric, namely baseline coverage probability, which is the (empirical) probability that the rank-sets $[\hat{l}(m), \hat{u}(m)]$ constructed by any method cover the rank-sets $[\tilde{l}(m), \tilde{u}(m)]$ constructed using the BASELINE method, *i.e.*,

$$\mathbb{P}\left(\bigcap_{m \in \mathcal{M}} \mathbf{1}\left\{[\tilde{l}(m), \tilde{u}(m)] \subseteq [\hat{l}(m), \hat{u}(m)]\right\}\right) \tag{15}$$

The baseline intersection probability, which we discussed in Section 5 and illustrated in Figure 1, is a less conservative metric compared to the baseline coverage probability. While the latter represents the probability that all rank-sets are covered by the BASELINE rank-sets, the former only considers the probability that the rank-sets intersect. Thus, achieving high baseline coverage probability is difficult, particularly when the BASELINE rank-sets are larger.

**Quality of the rank-sets when considering the baseline coverage probability.** In Figure 7, we show the average rank-set size against the baseline coverage probability for rank-sets constructed using all methods (except BASELINE) for $n = 990$ and different values of $\alpha$ (the same values as in Figure 6). Immediately, we notice that the baseline coverage probability of all methods is very low. For rank-sets constructed using pairwise comparisons only by a strong LLM (LLM GPT4, LLM CL3 and LLM GPT3.5), the baseline coverage probability is close to (or exactly) zero. Rank-sets constructed using only pairwise comparisons by humans (HUMAN ONLY) or prediction-powered ranking using a strong LLM (PPR GPT4, PPR CL3 and PPR GPT3.5) achieve better baseline coverage probability. However, the difference in performance among these methods is not clear to distinguish, which motivated us to consider the baseline intersection probability metric for our experimental results in Section 5.

In Figure 8, we show the average rank-set size against the baseline coverage probability for rank-sets constructed using PPR GPT4, PPR GPT3.5 and PPR CL3 for different values [9] of $n$ and $\alpha$ (the same $\alpha$ values as in Figure 6). Similarly as in Figure 2, results show that there is a trade-off between average rank-set size and the baseline coverage probability, which improves rapidly as the number of pairwise comparisons by humans $n$ increases, but with diminishing returns.

---

[8] $\alpha \in \{0.4, 0.3, 0.25, 0.2, 0.15, 0.1, 0.075, 0.05, 0.025, 0.01\}$

[9] $n \in \{66, 132, 198, 462, 990, 1452, 1980, 2442, 2970\}$.

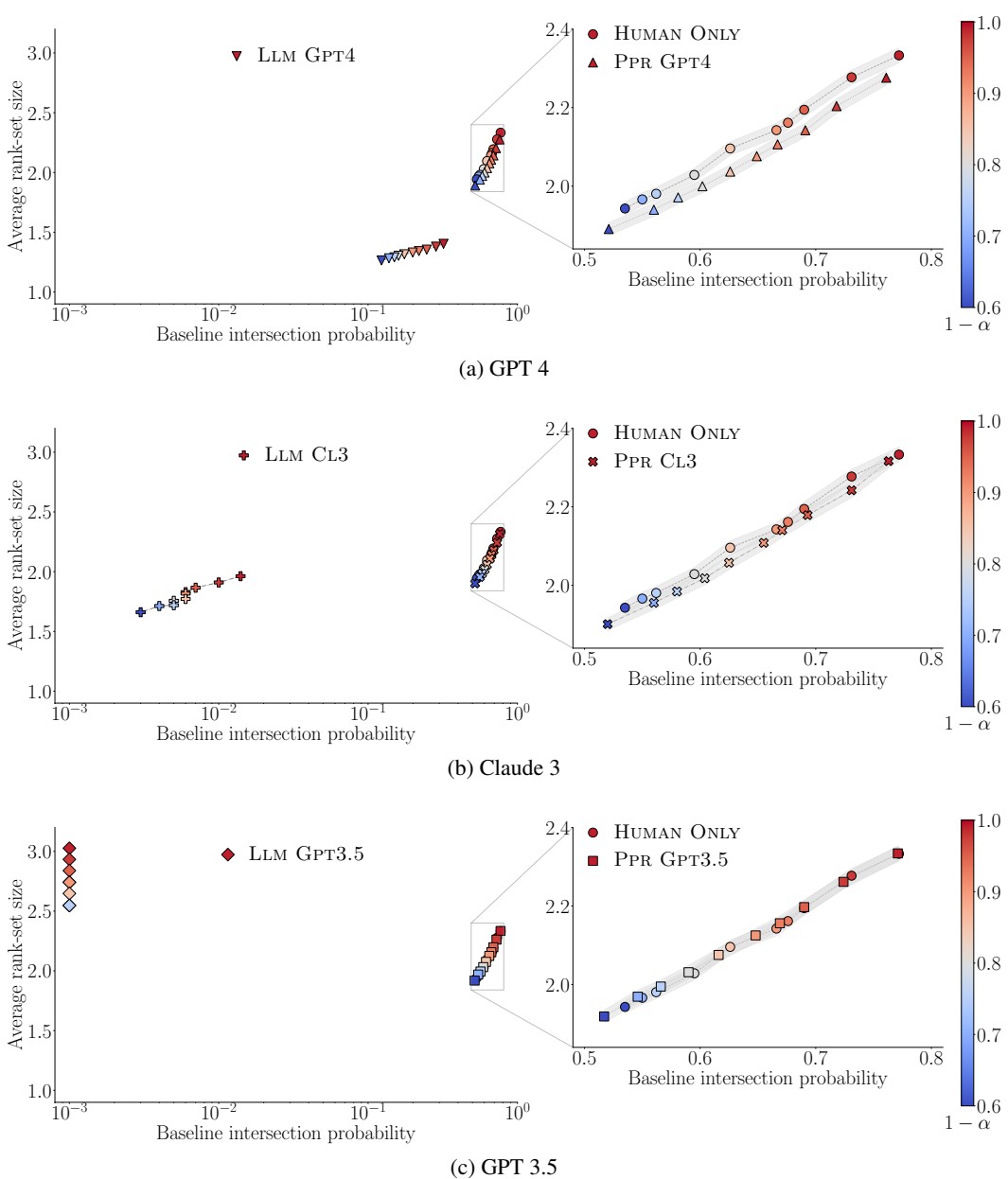

Figure 6: Average rank-set size against baseline intersection probability for rank-sets constructed using only pairwise comparisons by a strong LLM: LLM GPT4 (top), LLM CL3 (middle) and LLM GPT3.5 (bottom), only pairwise comparisons by humans (HUMAN ONLY), and pairwise comparisons by both a strong LLM and humans (PPR GPT4 top, PPR CL3 middle and PPR GPT3.5 bottom) for different $\alpha$ values and $n = 990$. Smaller (larger) average rank-set sizes and larger (smaller) intersection probabilities are better (worse). The shaded region shows a $95\%$ confidence interval for the rank-set size of all rank-sets among all $1,000$ repetitions.

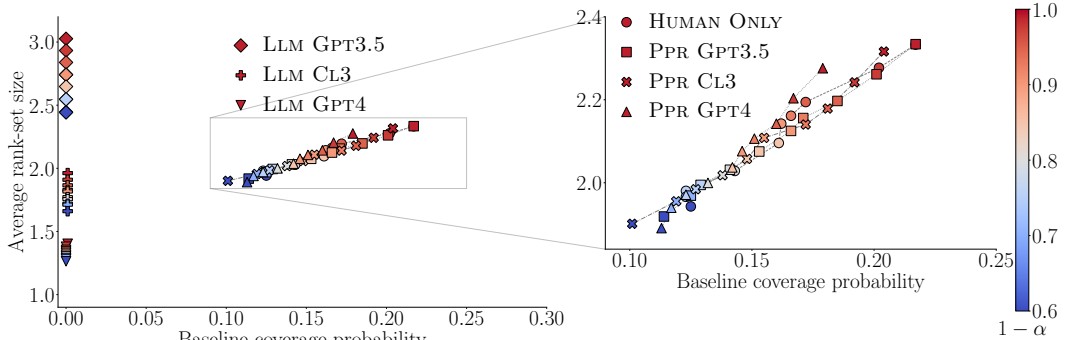

Figure 7: Average rank-set size against baseline coverage probability for rank-sets constructed using only pairwise comparisons by a strong LLM (LLM GPT4, LLM GPT3.5 and LLM CL3), only pairwise comparisons by humans (HUMAN ONLY), and pairwise comparisons by both a strong LLM and humans (PPR GPT4, PPR GPT3.5 and PPR CL3) for different $\alpha$ values and $n = 990$. Smaller (larger) average rank-set sizes and larger (smaller) coverage probabilities are better (worse). In all panels, 95% confidence bars for the rank-set size are not shown, as they are below 0.02.

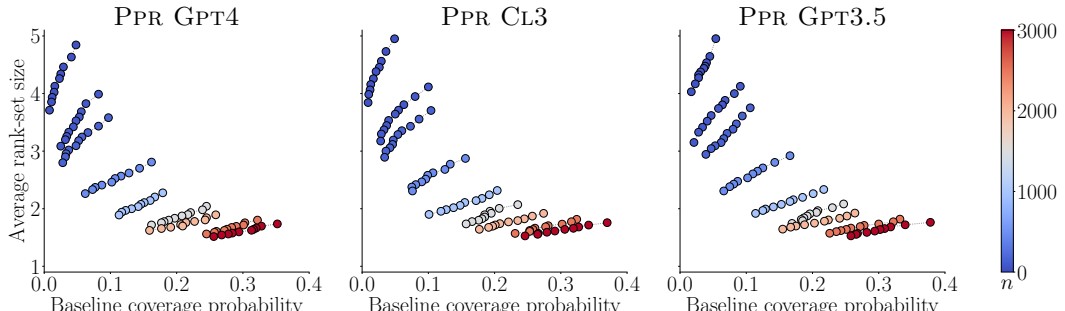

Figure 8: Average rank-set size against baseline coverage probability for rank-sets constructed using pairwise comparisons by both a strong LLM and humans for different $n$ and $\alpha$ values. Smaller (larger) average rank-set sizes and larger (smaller) coverage probabilities are better (worse). In all panels, 95% confidence bars for the rank-set size are not shown, as they are below 0.04.

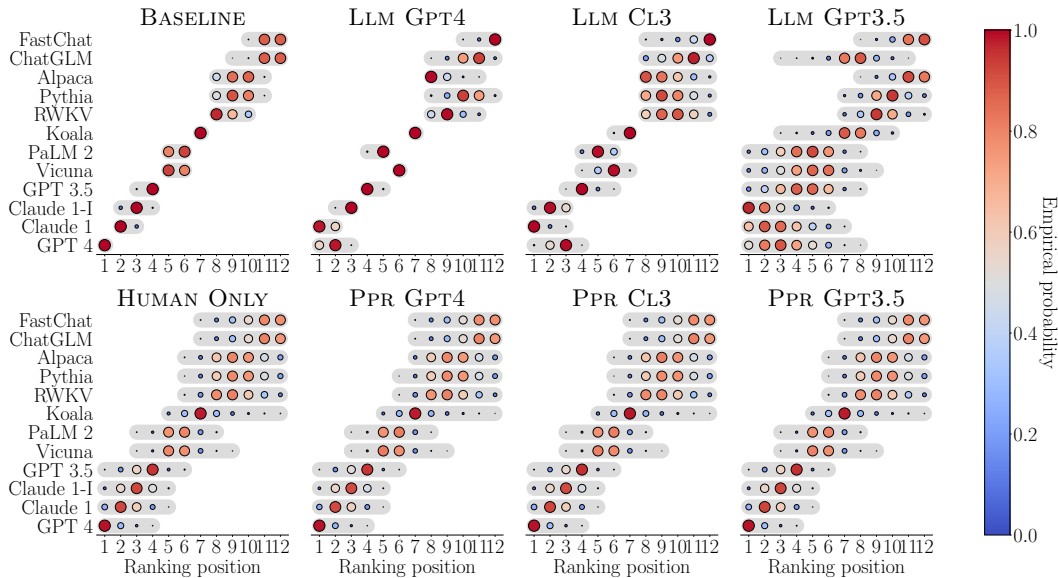

Figure 9: Empirical probability that each ranking position is included in the rank-sets constructed by all methods for each of the LLMs under comparison. In all panels, $n = 990$ and $\alpha = 0.05$. Larger (smaller) dots indicate higher (lower) empirical probability.

## D.3    Structure of the Rank-sets

In this subsection, we take closer look at the structure of the rank-sets constructed by all methods. We compute the empirical probability that each ranking position is included in the rank-sets constructed by all methods, of each of the LLMs under comparison. The results are summarized in Figure 9 for $\alpha = 0.05$ and $n = 990$.

**Empirical probability of ranking positions.** Our first observation is that, the ranking positions of each model in LLM GPT4 and LLM CL3 have lower uncertainty compared to PPR GPT3.5 and PPR CL3, respectively. However, LLM GPT3.5 exhibits higher uncertainty compared to PPR GPT3.5. Nonetheless, the ranking positions with the highest probability mass in LLM GPT4, LLM CL3 and LLM GPT3.5 significantly deviate from the BASELINE. Specifically, the ranking position with highest probability mass differs for 7 out of 12 models. In contrast, for PPR GPT4, PPR CL3 and PPR GPT3.5 it only differs from the BASELINE for 3 out of 12 LLMs. These findings once again question the rationale of relying solely on pairwise comparisons by strong LLMs to rank LLMs [12, 25–29, 31]. Our second observation is that there is no significant difference in the uncertainty in ranking positions across HUMAN ONLY, PPR GPT4, PPR CL3 and PPR GPT4. However, the distribution of probability mass across different ranking positions differs slightly among these methods. This observation is clearly seen in PPR GPT4, where the Alpaca model has zero probability mass for position 6.

**Empirical probability of rank-sets.** Next, we compute the empirical probability of each rank-set constructed by all methods and for each of the LLMs under comparison with $n = 990$ and $\alpha = 0.05$. The results are summarized in Figure 10. Consistent with the observations from Figure 9, we note that the distribution of rank-sets constructed by LLM GPT4 is more concentrated than those constructed by other methods. Conversely, LLM GPT3.5 exhibits a more spread-out distribution of rank-sets, indicating higher uncertainty in its ranking positions. This observation is consistent across all LLMs considered for ranking. Despite the more concentrated distributions of rank-sets for LLM GPT4 and LLM CL3, we observed that the ranking positions with the highest probability mass often differed from those of the BASELINE, with discrepancies observed in 7 out of 12 models. On the contrary, the rank-sets constructed by PPR GPT4, PPR CL3 and PPR GPT3.5 exhibit distributions that are neither excessively spread out nor highly concentrated. But the rank-sets with the highest probability mass constructed by these methods coincide with those constructed by BASELINE more frequently than their LLM only counterparts. Once again, these findings underscore our argument that rank-sets obtained using only pairwise comparisons of strong LLMs are not very reliable.

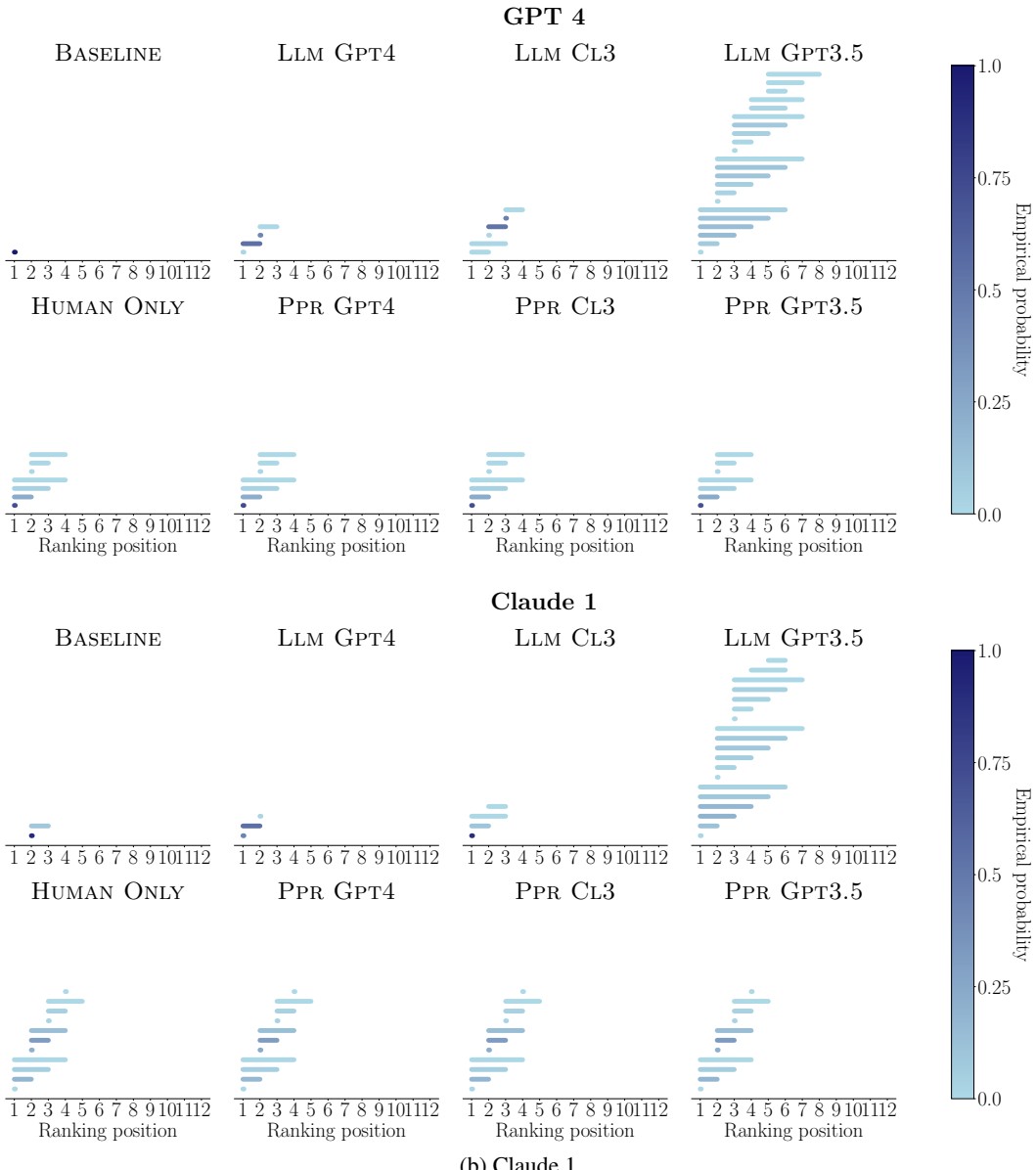

Figure 10: Empirical probability of each rank-set constructed by all methods for all 12 models (one model per sub-figure). In all panels, $n = 990$ and $\alpha = 0.05$.

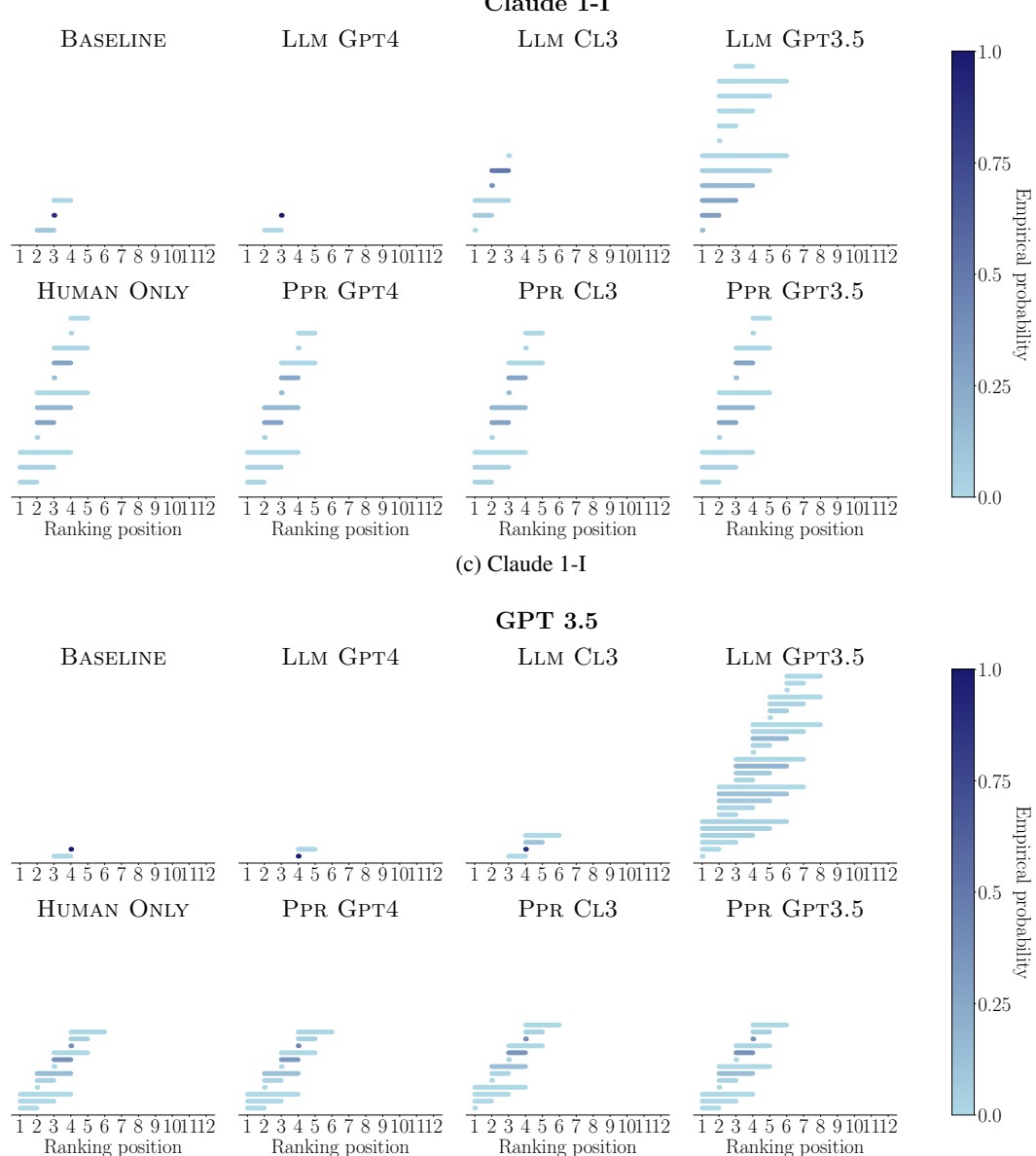

(c) Claude 1-I

Figure 10 (cont.)

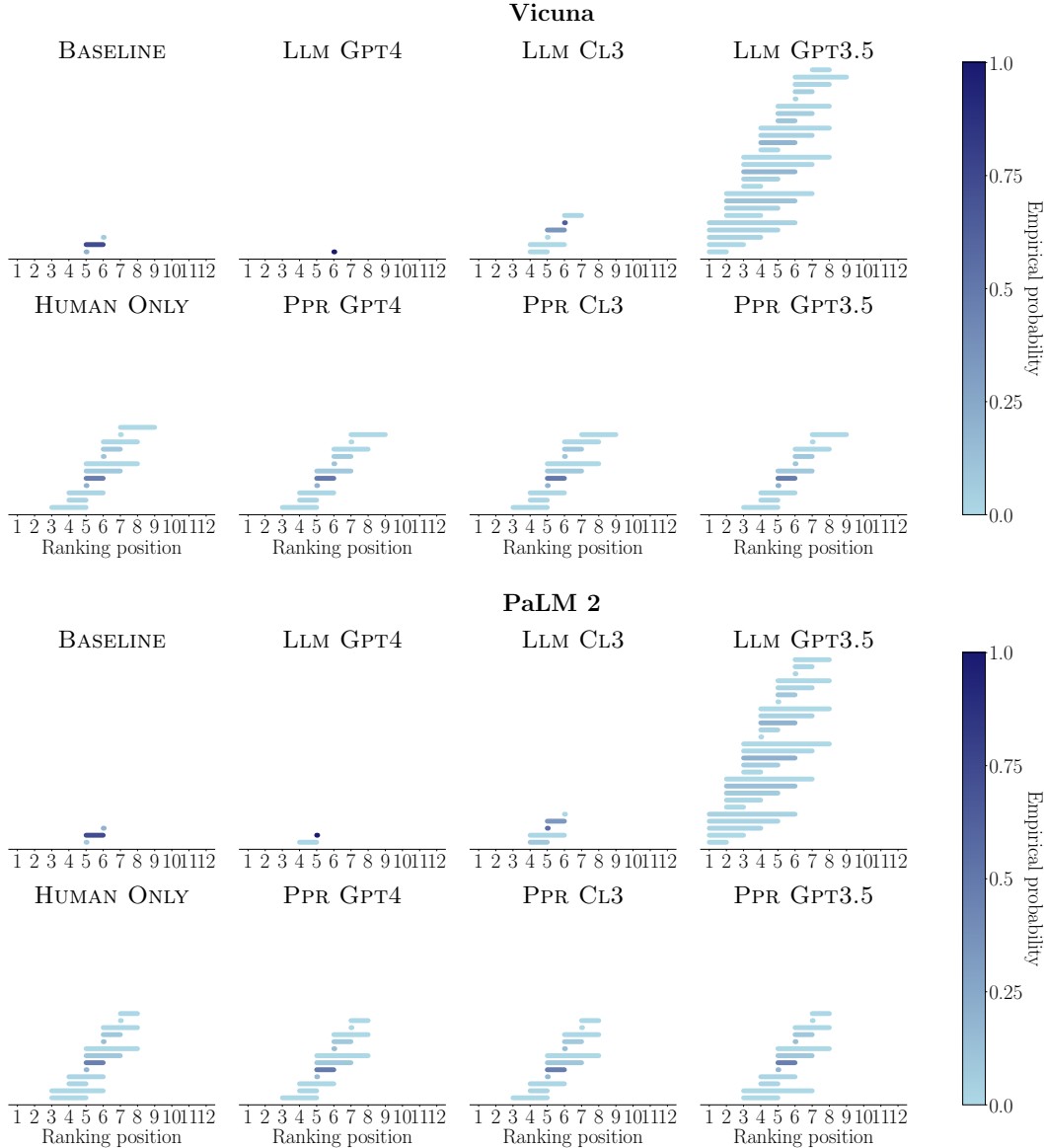

Figure 10 (cont.)

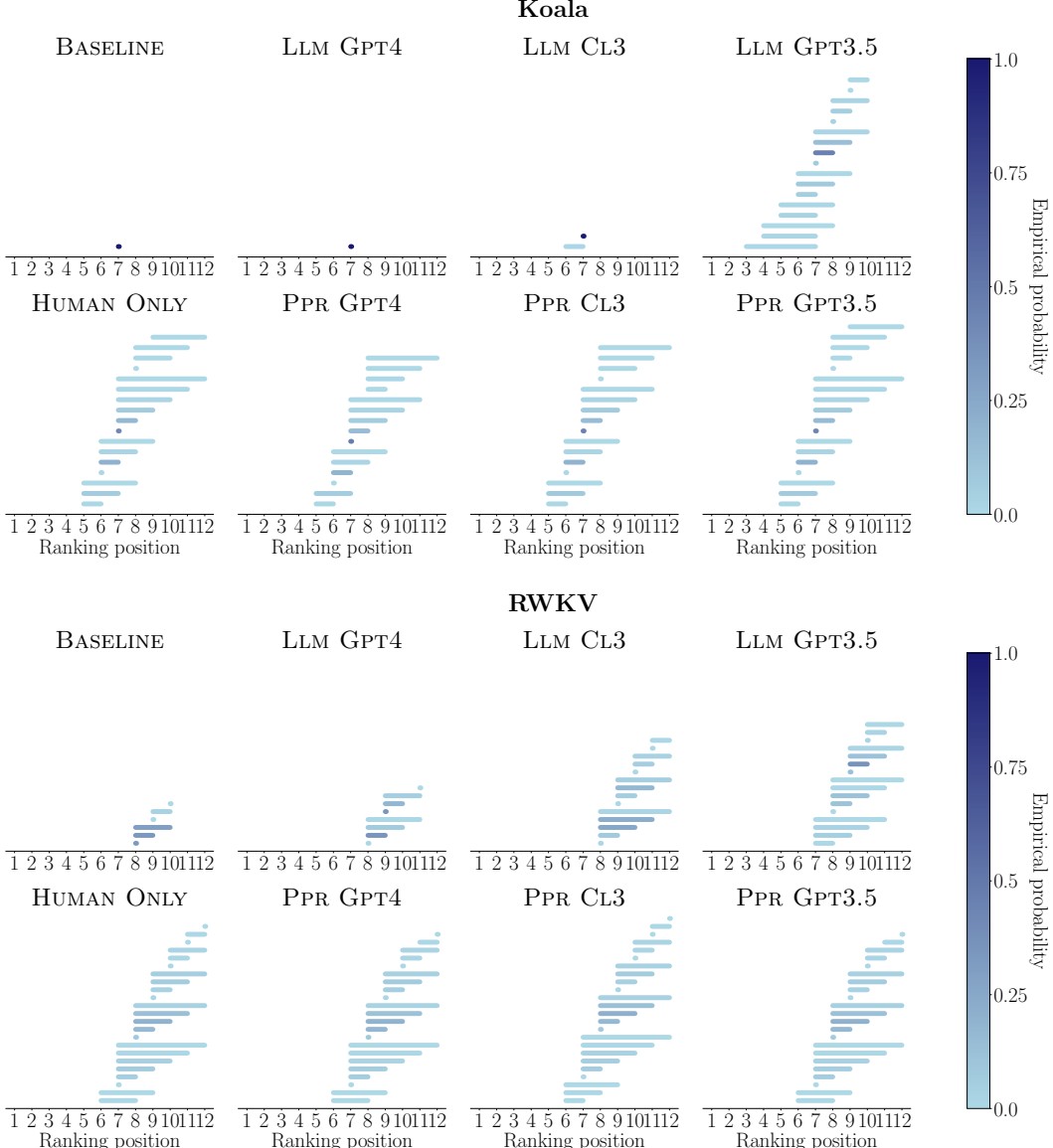

Figure 10 (cont.)

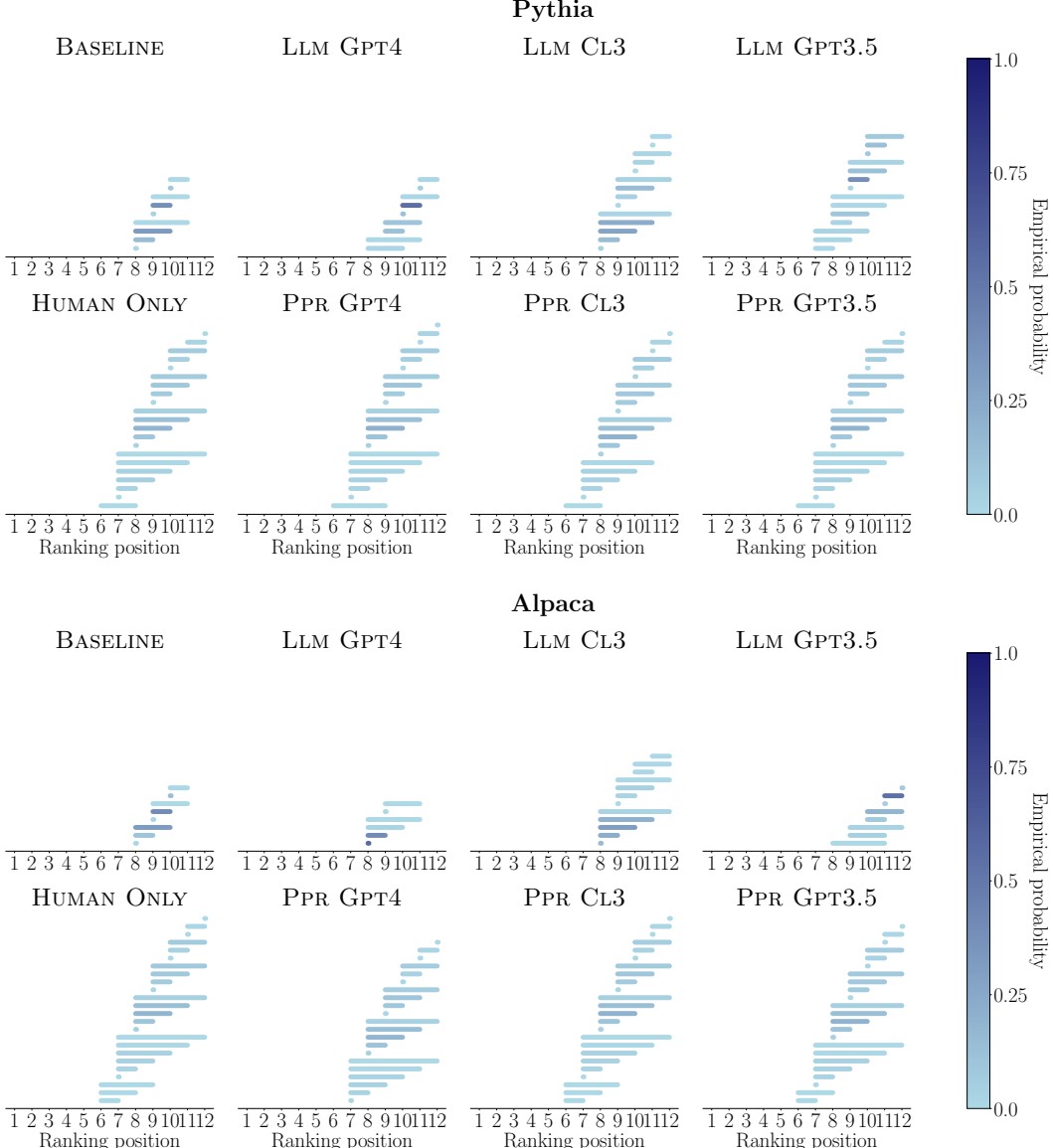

Figure 10 (cont.)

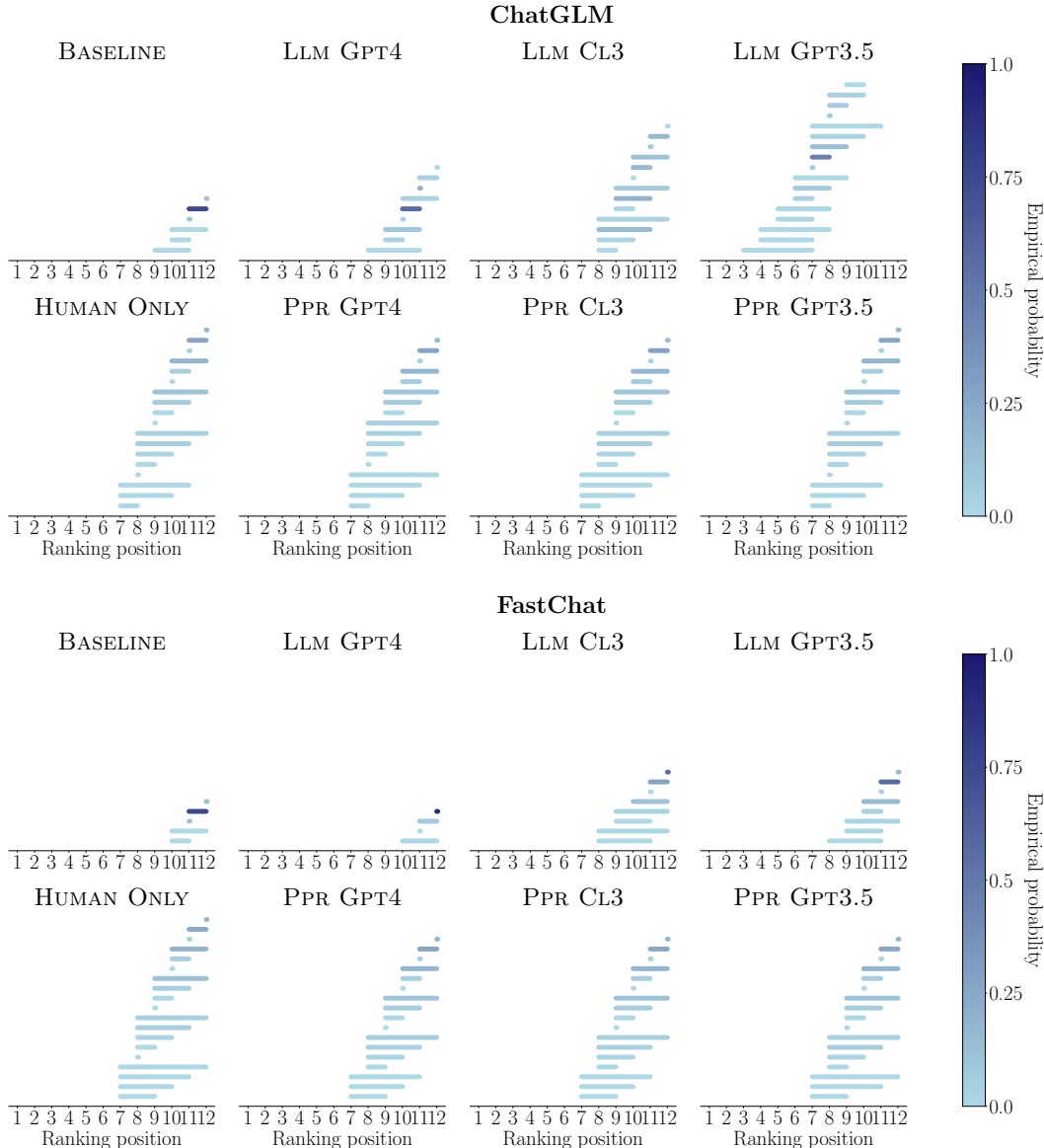

Figure 10 (cont.)

# E   Synthetic Experiments

In this section, we evaluate our framework in a synthetic setting where the true rank-sets of the models are known. This allows us to validate the theoretical coverage guarantee of Theorem 4.1 without relying on a proxy metric, and also compute the rank-biased overlap (RBO) [65].

**Experimental setup.** We consider $k = 8$ models and, in each experiment, we generate a random vector of true win probabilities $\boldsymbol{\theta}$ (Eq. 1), which induces the true rank-sets of the models.[10] To obtain win probabilities $\breve{\boldsymbol{\theta}}$ (Eq. 3), we add random noise to the vector and then re-normalize $\breve{\boldsymbol{\theta}}$. We sample the noise from a Uniform$(-u, u)$ distribution, where $u$ a parameter we manually set to simulate different alignment levels between strong LLMs and human preferences. A larger $u$ indicates a greater difference between $\boldsymbol{\theta}$ and $\breve{\boldsymbol{\theta}}$, meaning the LLM is less aligned with human preference, and vice versa. In our experiments, we set $u \in \{0.05, 0.1, 0.3\}$ to simulate three different strong LLMs.

To draw reliable conclusions, for each experiment, we create rank-sets 300 times, each time using a different set of $n + N = 50{,}000$ simulated pairwise comparisons by humans and the three strong LLMs, with an equal number of pairwise comparisons per pair of models. We ensure that each model provides the first and second response to an equal number of pairwise comparisons. Let $m_a$ and $m_b$ be the two models in a pairwise comparison, with $m_a$ giving the first response. For each pairwise comparison, first, we generate a number $x \in (0, 1)$ uniformly at random. For the human outcome, if $x < 2\theta_{m_a}$, then the response of model $m_a$ is preferred ($w = 1$). Similarly, for the strong LLM outcome, if $x < 2\breve{\theta}_{m_a}$, then the response of model $m_a$ is preferred ($\hat{w} = 1$). In every comparison we set $w' = \hat{w}' = 0$.

**Coverage probability.** Using the generated pairwise comparisons, we compute rank-sets in a similar way as described in Section 5, with $\alpha = 0.1$. Since the true rank-sets are known, we can compute the (empirical) coverage probability, shown in Figure 11. The results show that the coverage probability increases with $n$, consistent with Theorem 4.1. Further, the coverage probability is greater when $u$ is smaller, indicating that our method achieves better results when the strong LLM is more aligned to human preference.

**Rank-biased overlap (RBO).** For each method, we obtain a ranking $\hat{T}$ by sorting the models in descending order of their $\hat{\boldsymbol{\theta}}$ values. We then compute the RBO metric relative to their true ranking $T$ as follows:

$$\text{RBO}(T, \hat{T}, p) = (1 - p) \sum_{i \in [k]} p^{i-1} \frac{|T_{:i} \cap \hat{T}_{:i}|}{i}$$

where $T_{:i}$ and $\hat{T}_{:i}$ represents the top $i$ models in ranking $T$ and $\hat{T}$, respectively, and $p \in [0, 1]$ is a chosen parameter. When $p = 1$, all models are weighed equally. As we decrease $p$, more emphasis is given to the top-ranked models, and at $p = 0$, only the top-ranked model is considered. In Figure 12, we compare RBO values as we increase the number of human pairwise comparisons $n$, for $p = 0.6$. The results show that increasing $n$ improves RBO across all methods. Additionally, combining human pairwise preferences with a strong LLM further improves RBO values, demonstrating that our method performs better than those solely relying on strong LLM preferences. We repeated our experiments with multiple values of $p \in [0, 1]$ and observed no significant variation in the results.

---

[10]In our experiments, we generate true win probabilities $\boldsymbol{\theta}$ with unique values, so the rank-sets are always singletons, resulting in a unique true rank for each model.

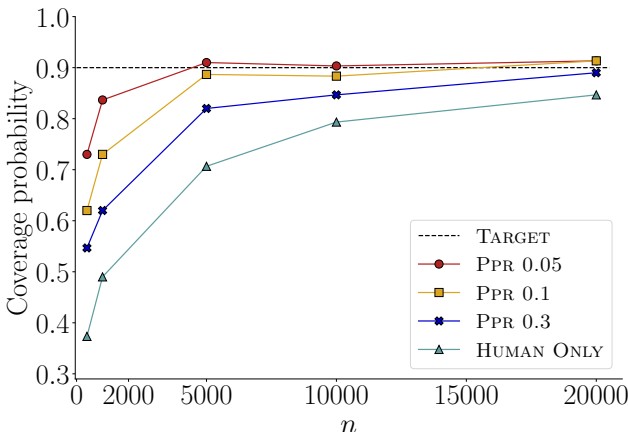

Figure 11: Empirical coverage probability of the rank-sets constructed using only $n$ synthetic pairwise comparisons by humans (HUMAN ONLY) and using both $n$ synthetic pairwise comparisons by humans and $N + n$ synthetic pairwise comparisons by one out of three different simulated strong LLMs (PPR 0.05, PPR 0.1 and PPR 0.3) with $\alpha = 0.1$ and $N + n = 50000$. Each of the strong LLMs has a different level of alignment with human preferences controlled by a noise value $u \in \{0.05, 0.1, 0.3\}$. The dashed line indicates the $1 - \alpha$ target coverage. The empirical coverage of the rank-sets constructed using only $N + n$ synthetic pairwise comparison by one of the same three strong LLMs (not shown in the figure) is 0.38 ($u = 0.05$), 0.13 ($u = 0.1$) and 0.0 ($u = 0.3$).

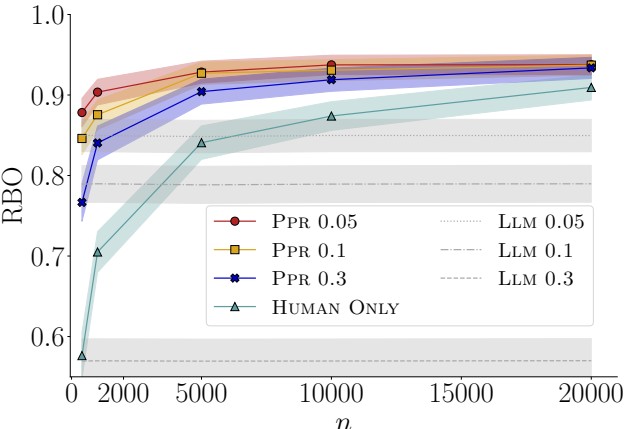

Figure 12: Average rank-biased overlap (RBO) of rankings constructed by ordering the empirical win probabilities $\hat{\boldsymbol{\theta}}$ estimated using only $N + n$ synthetic pairwise comparisons by one out of three different simulated strong LLMs (LLM 0.05, LLM 0.1 and LLM 0.3), only $n$ synthetic pairwise comparisons by humans (HUMAN ONLY), and both $n$ synthetic pairwise comparisons by humans and $N + n$ synthetic pairwise comparisons by one out of the same three strong LLMs (PPR 0.05, PPR 0.1 and PPR 0.3) for $\alpha = 0.1$ and $N + n = 50000$. Each of the strong LLMs has a different level of alignment with human preferences controlled by a noise value $u \in \{0.05, 0.1, 0.3\}$. RBO was computed with respect to the true ranking constructed by ordering the true win probabilities $\boldsymbol{\theta}$, for $p = 0.6$. The shaded region shows a 95% confidence interval for the RBO among all 300 repetitions.

