# OpenReview forum: "Prediction-Powered Ranking of Large Language Models"
_NeurIPS.cc/2024/Conference — NeurIPS 2024 poster_

### Official Review · Reviewer_Fevo · 2024-07-08

**Soundness:** 3
**Presentation:** 2
**Contribution:** 3
**Rating:** 6
**Confidence:** 3

**Summary:**

The paper studies uncertainty estimate in the LLM ranking problem, where the task is to rank LLMs based on their response quality. Ideally the labels should be from humans, but due to the cose, people use models such as GPT-4 as auto raters. There lacks a good study of uncertainty estimation in the problem. The paper applies prediction powered inference (PPI) to construct a rank set for each candidate LLMs. Experiments are conducted on the Chatbot Arena data. It is shown that the PPI approach work as intended to produce reasonable rank set size vs accuracy (when comparing with an oracle method using only human data) trade-offs.

**Strengths:**

The application of PPI to constructing rank set from pairwise comparisons from LLM evaluators is an interesting and timely application to the reviewer.

The proposed algorithms look sound by following PPI.

Overall the experiments demonstrate some desired behaviors of the proposed approach. Some analysis, such as the structure of the rank-sets, are interesting.

**Weaknesses:**

Overall I am positive of the paper as I feel the problem is important and applying PPI is a good proposal. However, the reviewer is not enthusiastic enough to give a higher rating due to the following concerns:

The paper is a more or less straightforward application of PPI to rank set construction. The theoretical properties shown in this paper mostly strictly follow those of PPI. Thus, the depth of this work is not substantial enough to warrant a higher rating in terms of novelty and technical depth.

The rank set is at dataset level, which may not be very useful in practice. For example, compared to a standard usage of PPI on a numeric metric, which will provide a concrete internal. For the paper, people end up with a discrete rank set for each candidate model. While it provides some uncertainty by looking at the set size, one may still end up wondering how useful that is. For example, if model 1 has rank 2,3,4, and model 2 has rank 3,4,5. It provides some information that model 1 seems better, but the reviewer is not sure how useful it really is (e.g., by how much? - as the guarantee is at the set level).

There are several points about the experimentation that the reviewer is not certain about
- As the authors acknowledged, only one dataset is used so the generalization is less clear. F
- The baseline / ground truth still needs some processing, such as a regression fit. This is different from standard tasks where human ground truth are given without the need to process anything. Thus the reviewer is not fully convinced how solid the conclusions are, e.g., “questioning the rationals used an extensive line of work that … (use LLM rankings for evaluation)”
- Figure 1 is not entirely convincing - at least the PRP methods do not achieve pareto frontier here. The reviewer understands the argument about the x axis, still, for a pareto problem, one may not really argue one method is better than the other if it is not pareto optimal.

The flow of the paper may be improved. The reviewer was puzzled about the methods and algorithm 1,2,3 when reaching to the experiments. Figure 1 is not very easy to interpret.

Some other minor limitations / future work, some are discussed in the paper: iid assumption - in practice, there could be a bias, e.g. using active learning to send data to human.

**Questions:**

See above.

**Limitations:**

The authors list several concrete limitations. Most are treated as future work.

---

> ### Author Rebuttal · Authors · 2024-08-05
>
> **[Application of PPI]** To the best of our knowledge, existing work has always applied PPI to construct confidence intervals for numerical quantities. In contrast, our work is the first to apply PPI to construct rank-sets and does so in a very timely domain, LLM evaluation.
>
> **[Rank set vs numeric metric]** We would like to first point out that rank-sets have been used in the literature as measures of uncertainty in rankings (see, for example, [48, 50]). In our setting, we believe rank-sets may provide useful information—the rank-sets tell the practitioner to what extent the difference in win-rates across models, after taking into account the uncertainty summarized by the confidence ellipsoid, are sufficient for each model to be ranked above or below other models. That being said, a practitioner may decide to use our framework to obtain not only estimates of the rank-sets but also estimates of the win-rates, which is a numeric metric, and the confidence ellipsoid used to construct the estimates of the rank-sets. We will clarify this in the revised version of the paper.
>
> **[One dataset / generalization]** Since our computational framework is rather generic, it is true that it could be readily applied to other benchmarks. However, this would require significant funding and time and, given the on-going discussions about the reliability of LLM evaluation, we believe the NeurIPS community may benefit more from our computational framework if published early. Further, given the rapid development of both new LLMs and benchmarks, we also feel that any claim of superiority of a LLM over others based on the estimated rank-sets on (static) benchmark data may be quickly outdated and thus have limited value.
>
> **[Baselines]** Since the regression fit is just an empirical estimate of a set of means (the win-rates), we do not find any reason to raise doubts about the baseline. Nevertheless, during the rebuttal period, we have carried out an additional evaluation of our framework in a synthetic setting. Since in the synthetic setting, the ground truth ranking is known and given, we do not need to rely on any processing to validate the theoretical coverage guarantees. Refer to the (general) author rebuttal and attached pdf for more details. We will include this additional evaluation in an Appendix in the revised version of the paper.
>
> **[Pareto frontier]** In the conclusions we have drawn from Figure 1 in lines 264-278, we do not (mean to) claim that the PPR methods achieve pareto frontier against all baselines. In lines 266-272, we claim that, **in terms of baseline intersection probability**, PPR methods are better than baseline methods using only (the same number of) comparisons by strong LLMs. In lines 273-278, we claim that PPR methods indeed achieve pareto frontier against a baseline method (HUMAN ONLY) using the same number of human comparisons as PPR methods since they are better both in terms of baseline intersection probability and size of the rank-sets. To avoid any misunderstanding, in the revised version of the paper, we will explicitly highlight that the baseline methods using only comparisons by strong LLMs GPT 4 and Claude 3 return smaller rank-sets than the PPR methods.
>
> **[Flow of the paper]** Following the reviewer’s suggestion, we will improve the flow of the paper and the description of Figure 1 in the revised version of the paper.
>
> **[Limitations]** Following the reviewer's suggestion, we will expand the discussions of the limitations and highlight that, in practice, active learning may be used to gather human pairwise comparisons.

---

### Official Review · Reviewer_EPH6 · 2024-07-10

**Soundness:** 3
**Presentation:** 3
**Contribution:** 3
**Rating:** 6
**Confidence:** 3

**Summary:**

This paper proposes a statistical framework to rank a collection of LLMs according to how well their output aligns with human preferences. This framework does this using a small set of human-obtained pairwise comparisons from LMSYS Chatbot Arena platform and a larger set of pairwise comparisons by a "strong" LLM and additionally provides an uncertainty estimate by giving a set of rankings for each LLM being compared. This study shows that, with at least probability threshold the user can set, the predicted rankings will eventually become increasingly likely to match the true order in which humans would prefer the models. The authors perform several experiments to empirically demonstrate the valitdity of the proposed framework.

**Strengths:**

- The framework is clearly explained and the paper is easy to follow
- The paper studies an interesting problem that focuses on ranking LLM in the context of scarcity of gathered pairwise comparisons by humans
- The empirical evaluation is thorough

**Weaknesses:**

- unless i misunderstood something, the small set of human pairwise comparisons has length = 1. Although, the potential bias and truthfulness of the human pairwise comparisons has been discussed in the limitation section, i think that it could be interesting to explore the potential error propagation in the rank sets from erroneous human comparisons.
- it is not clear to me, how the self-recognition [1] problem can be tackled with this framework. It has been shown that LLMs have non trivial capability of recognizing their own generation. Would this not be the case for one of the strong LLMs? wouldn't they tend to rank their generation higher? This fact combined with the previous question might affect the generazalization ability of this framework.


[1]: Panickssery, Arjun, Samuel R. Bowman, and Shi Feng. "Llm evaluators recognize and favor their own generations." arXiv preprint arXiv:2404.13076 (2024).

**Questions:**

see weaknesses.

**Limitations:**

limitations have been discussed.

---

> ### Author Rebuttal · Authors · 2024-08-05
>
> **[Small set of human pairwise comparisons]** The small set of human pairwise comparisons has length = $n > 1$ and we evaluate the performance of our computational framework for different values of $n$ in Figure 2.
>
> **[Erroneous human comparisons]** We agree that it would be interesting to explore the potential error propagation in the rank sets from erroneous human comparisons due to, e.g., bias, lack of truthfulness or strategic behavior, however, as pointed out in the limitation section, this is left as future work.
>
> **[Self-recognition problem]** If the strong LLM tends to rank its generation higher due to self-recognition, this is corrected by our method, as PPR accounts for possible biases of the strong LLM using the small set of $n$ pairwise comparisons by humans. In fact, in our experiments, Claude 3 suffers from the self-recognition problem and our framework corrects for such bias. More specifically, in Figure 9 in Appendix D.3, LLM CL3 prefers to rank the two Claude 1 models higher than the baseline but PPR CL3 corrects for this bias.

---

> > ### Comment · Reviewer_EPH6 · 2024-08-12
> > **Response to rebuttal**
> >
> > I appreciate the authors' responses to my questions and additional clarifications. Overall, I would like to keep the current score rating.

---

### Official Review · Reviewer_G9Nw · 2024-07-12

**Soundness:** 4
**Presentation:** 3
**Contribution:** 3
**Rating:** 7
**Confidence:** 3

**Summary:**

- Focuses on uncertainty in rankings using a small set of human pairwise comparisons and a large set of model estimated comparisons using a concept of rank sets. A rank set is a set of ranks a specific model can take. A large rank set indicates high uncertainty in ranking position and vice-versa a small set implies a confident rank assessment. The method works by constructing a confidence ellipsoid which in turn using methods of prediction powered inferences (methods to estimate confidence intervals when you have a small set of labeled gold standard data and a large set of machine labeled data). The paper employs the methods to rank 12 LLMs using data from LMSYS Chatbot Arena.

**Strengths:**

- Evaluations of LLMs is particularly challenging and estimating ranking of models for specific tasks an important area. This paper makes a good contribution towards this by investigating ranking uncertainty using prediction powered inferences. The overall setting (small set of human-annotated data and a large machine labeled dataset) is realistic and therefore the work lends itself to practical use as well.

- The methodological contributions are interesting in itself and the concept of using rank sets to characterize uncertainty intriguing.

- The exposition and presentation of material is good. Some of the plots are well structured and intuitive to grasp (Figure 3 in particular is well crafted).

**Weaknesses:**

- The main weakness is on evaluation. The paper proposes two metrics: rank-set size and baseline intersection probability. Small rank-set sizes are better - presumably as the confidence in ranks is better, and baseline intersection probability - large is better as it supposes the baseline method is closer to the true ranking. First: The paper should really report more standard ranking metrics such as precision/recall @ k, RBO, MAP or Normalized Discounted Cumulative Gain (NDCG). To factor in rank sets, you could use most probable ranking from your rank sets to compute these. Second: the absence of true rankings makes the empirical aspects less convincing. A synthetic experiment where true rankings are known perhaps would make sense to empirically demonstrate the main claims.

- Use of machine labels just appear to widen the confidence bounds (e.g. Figure 3). The base conclusions on ranking appears to remain the same. In this instance, therefore, it remains unclear what the value of machine labeled information is.

- Some additional insights into rank sets would add to the paper. For instance, how stable are rank sets to minor changes in $\mathcal{M}$? Uncertainty in rank for one model depends the overall set of models being considered, so this would be interesting to study. There are possibly other facets, such as how rank sets can be used in practice or the quality of the machine labeled data.

**Questions:**

NA

**Limitations:**

Yes.

---

> ### Author Rebuttal · Authors · 2024-08-05
>
> **[Evaluation metrics]** In our work, we focus on rank-sets as a measure of uncertainty in rankings and thus our experimental evaluation aims to assess the quality of the rank-sets estimated using our method and several baselines. In this context, we think that the ranking metrics proposed by the reviewer provide little information about the quality of the rank-sets. More importantly, we are unsure how to operationalize them in our setting. On the one hand, precision/recall @ k, MAP and NDCG are used in settings in which there is a set of items to be ranked and, for each item, one can measure whether an item is relevant or is not relevant. However, in our setting, it is difficult to measure whether a model is relevant or is not relevant based on win-rates. On the other hand, RBO is used in settings in which there is a ground-truth ranking. However, in our setting, we do not have access to a ground-truth ranking.
>
> That being said, following up on the reviewer's comment, we have carried out an additional evaluation of our framework in a synthetic setting where the ground-truth ranking is known. Then, in this synthetic setting, we have computed RBO using the most probable ranking, as suggested by the reviewer, as well as the empirical coverage. Refer to the (general) author rebuttal and attached pdf for more details. We will include this additional evaluation in an Appendix in the revised version of the paper.
>
> **[Synthetic experiments with true rankings]** As discussed in the previous response, during the rebuttal period, we have carried out an additional evaluation of our framework in a synthetic setting. Since in the synthetic setting, the true ranking is known, we have been able to validate the theoretical coverage guarantees without using a baseline. Refer to the (general) author rebuttal and attached pdf for more details. We will include this additional evaluation in an Appendix in the revised version of the paper.
>
> **[Value of machine labeled information in Figure 3]** In Figure 3, note that, by using machine labeled information, PPR GPT4 allows us to draw the same conclusions as BASELINE using significantly fewer human comparisons ($n$ vs. $N+n$). In Figure 1, the value of the machine labeled information is perhaps more apparent. Therein, the results show that PPR GTP4 achieves narrower confidence bounds (y axis) and higher baseline intersection probability (x axis) than HUMAN ONLY, a baseline that uses the same number of human comparisons but no machine comparisons.
>
> **[Additional insights]** In our work, we do not aim to make a comprehensive empirical evaluation of rank-sets as an uncertainty measure or the quality of machine labeled data. Therefore, we leave the study of the sensitivity of rank-sets to minor changes in $\mathcal{M}$, practical uses of rank-sets, and the quality of machine label data as interesting venues for future research.

---

### Official Review · Reviewer_dge6 · 2024-07-13

**Soundness:** 3
**Presentation:** 3
**Contribution:** 2
**Rating:** 6
**Confidence:** 4

**Summary:**

The paper tackles an interesting problem of evaluating ranking large language models automatically using a strong LLM as alternative to human preference estimates. The work primarily focuses on modelling uncertainty in such a ranking generated when compared to the distribution of human preference rankings. Since, pairwise comparison by humans are cumbersome and pairwise comparisons by strong LLMs are not completely consistent with human preferences, the authors propose a framework that improves upon pairwise ranking by strong LLMs. The authors propose a prediction power inference based framework to construct rank sets that provide coverage guarantees with respect to the true ranking consistent with human preferences.

**Strengths:**

1.	The work tackles an important problem concerning evaluation of ranking LLMs in an automated manner with respect to limited human preferences. The work discusses in details the drawbacks of existing ranking approaches and provides a statistically grounded framework (prediction powered inference) that works well in face of scarcity in human preference annotations.
2.	The authors perform extensive evaluation on chatbot arena and propose two measures, namely rank set size and baseline intersection probability.
3.	The proposed framework is useful for modelling uncertainty when using LLMs as judges and can also be applied to other scenarios such as modelling uncertainty in LLM  driven relevance judgements for offline evaluation of retrieval.

**Weaknesses:**

1.	While the authors perform extensive evaluation, it might be a good idea to also test on other benchmarks like MT-bench or AlpacaEval related to approximation of human judgements. While authors already discuss the generalization aspect in limitations with regards to this, I would like to add it would also help address the concern regarding selection bias of test set and evaluators. Additionally, due to input limitations the benchmark may also not be representative of tasks that require reasoning over long form inputs and specifically complex reasoning tasks. Hence the leaderboard may only weakly correlate with real-world performance on these tasks. Additionally a minor point is that the metrics considered for rating response “relevance , helpfulness , accuracy , creativity and level” as shown in prompt might also change depending on the task: For instance when evaluating on a benchmark akin to QA tasks where precise information is needed creativity may not be a valid metric anymore. Hence evaluating on more benchmarks would help give a clearer picture on usefulness of the proposed framework.

2.	While it is appreciated that the work provides the proof for theoretical coverage guarantees, the claim regarding the coverage guarantees made in Introduction and beginning of section 4 should be revisited as true rank-sets (true probabilities unknown) cannot be computed for LLMs. The Baseline intersection probability is a weak approximation for coverage guarantees. Though the authors argue that baseline method approximates well the true rank sets due to being constructed from large number of human pairwise comparisons this might not necessarily hold due to selection bias, distribution shift and various other factors.  Without further evidence the claim that  Baseline intersection probability is a good approximation for true coverage measure is not well supported.

**Questions:**

Did the authors also try few-shot prompting the strong LLMs by showing few examples on how to judge the responses ? Would be interesting to see if this leads to any change in final observation and insights.

For a small sample set would it be possible to empirically test coverage guarantees without the other metric being the proxy assumption?

**Limitations:**

Due to the inherent limitations of the benchmark, this work may also not be representative of tasks that require reasoning over long form inputs and specifically complex reasoning tasks.

---

> ### Author Rebuttal · Authors · 2024-08-05
>
> **[Other benchmarks]** Since our computational framework is rather generic, it is true that it could be readily applied to other benchmarks. However, this would require significant funding and time, and given the on-going discussions about the reliability of LLM evaluation, we believe the NeurIPS community may benefit more from our computational framework if published early. Further, given the rapid development of both new LLMs and benchmarks,  we also feel that any claim of superiority of an LLM over others based on the estimated rank-sets on (static) benchmark data may be quickly outdated and thus have limited value. Under **[Baseline method]**, we discuss selection bias of the test set and the evaluators.
>
> **[Reasoning tasks]** The goal of our experiments is to showcase and validate our computational framework, and not to make a comprehensive evaluation of LLMs across different tasks. Therefore, we do not claim that the conclusions derived from the rank-sets estimated using data from the LMSYS Chatbot Arena platform correlate with real-world performance on reasoning over long form inputs or complex reasoning tasks. We will clarify this in the revised version of the paper.
>
> **[Metrics and tasks]** We agree with the reviewer that each type of task may need to be evaluated using a different set of metrics. However, since our computational framework is generic and does not make any assumption about the metrics considered for rating each response, we do not find any reason for our framework not to be applicable. That said, as discussed under **[Other benchmarks]**, conducting experiments on other tasks and/or benchmarks would require significant funding and time and, given the on-going discussions about the reliability of LLM evaluation, we believe the NeurIPS community may benefit more from our computational framework if published early.
>
> **[Baseline method]** We would like to clarify that our claim is that the baseline intersection probability is a reasonable proxy for the coverage with respect to the true rank-sets induced by the **distribution of queries used by users at LMSYS** and the **distribution of human preferences of the users at LMSYS**. In this context, we would further like to clarify that, in Appendix D.2, we have also verified that, using a more conservative baseline metric, our conclusions also hold.
>
> Relatedly, we do acknowledge that, if the users at LMSYS are not representative of the target query distribution and distribution of human preferences, the estimated rank-sets both by our method and the baseline method may be inaccurate (due to selection bias and distribution shift). However, studying such a setting is left as future work. We will clarify this in the discussion of the limitations in the revised version of the paper.
>
> **[Few-shot prompting]** We did not try few-shot prompting of the strong LLMs but instead used (almost) the same prompt used by Zheng et al. [12]. We agree with the reviewer that it would be interesting to investigate and optimize the type of prompting used to elicit pairwise comparisons by strong LLMs. However, this is a research question on its own and is also left as future work.
>
> **[Empirically test coverage guarantees]** During the rebuttal period, we have carried out an additional evaluation of our framework in a synthetic setting. Since in the synthetic setting, the true ranking is known, we have been able to validate the theoretical coverage guarantees without using a proxy metric. Refer to the (general) author rebuttal and attached pdf for more details. We will include this additional evaluation in an Appendix in the revised version of the paper.

---

### Author Rebuttal · Authors · 2024-08-05

We would like to thank the reviewers for their careful and insightful comments, which will help improve our paper. We include point-by-point responses to each reviewer in individual rebuttals. Moreover, in what follows, we provide details of an additional evaluation of our framework using a synthetic setting, which we have conducted during the rebuttal period. We refer to this evaluation in responses to three of the reviewers in the individual rebuttals (dge6, G9Nw, Fevo). The results of this evaluation are attached as a one page pdf.

Since in the synthetic setting, the true ranking is known, we have been able to validate the theoretical coverage guarantees without using a proxy metric, and we have also computed rank-based overlap (RBO) using the most probable ranking, as suggested by reviewer G9Nw. We will include this additional evaluation in an Appendix in the revised version of the paper. In the following paragraphs, we elaborate on this synthetic experimentation.

Initially, we set the number of models to $k=8$. In each experiment, we generated a random vector of true win probabilities $\boldsymbol{\theta}$ (Eq. 1), which induces a true ranking of the models. To generate the win probabilities $\boldsymbol{\breve{\theta}}$ (Eq. 3), we added random noise to the vector $\boldsymbol{\theta}$, then re-normalized $\boldsymbol{\breve{\theta}}$. The noise was sampled from a $Uniform(-u,u)$ distribution, where $u \in (0,1)$ a parameter we manually set to simulate different alignment levels of strong LLMs to human preference. Intuitively, the larger the value of $u$, the larger the difference between $\boldsymbol{\theta}$ and $\boldsymbol{\breve{\theta}}$, and the less aligned the strong LLM is to human preference. In our experiments, we set $u=\\{0.05, 0.1, 0.3\\}$, simulating three different strong LLMs.

To draw reliable conclusions for each experiment, we created rank-sets $300$ times, each time using a different set of $N+n=50 000$ simulated pairwise comparisons by humans and the three strong LLMs, with an equal number of pairwise comparisons per pair of models.

Let $m_a$ and $m_b$ be the two models participating in a pairwise comparison, with $m_a$ being the model that gave the first response. We ensure that each model provides the first and second response to an equal number of pairwise comparisons. For each pairwise comparison, we generate uniformly at random a number $x \in (0,1)$.  For the human outcome, if $x < 2 \theta_{m_a}$ then the response of model $m_a$ is preferred ($w=1$). Similarly, for the strong LLM outcome, if $x < 2 \breve{\theta}_{m_a}$ then the response of model $m_a$ is preferred ($\hat{w}=1$). In every comparison we set $w’ = \hat{w}' = 0$.

From the generated pairwise comparisons, we computed rank-sets in a similar manner as section 5 in our paper, using $\alpha=0.1$. Since the true ranking is known, we can compute the coverage, shown in Figure 1 in the attached pdf. The coverage increases with $n$, in agreement with our theoretical result in Theorem 4.1.

By sorting the models in descending order of the estimated $\boldsymbol{\hat{\theta}}$, we obtain the most probable ranking of each method and can compute the RBO of each method’s ranking with respect to the true ranking, shown in Figure 2 in the attached pdf. We can see that combining pairwise comparisons of humans and a strong LLM results in the highest RBO values.

---

### Decision · Program_Chairs · 2024-09-25

**Decision:**

Accept (poster)

**Comment:**

This paper studies how to rank Large Language Models (LLMs) automatically, using a strong LLM as an alternative to human preferences. It achieves this by focusing on the uncertainty that arises in such rankings compared to the distribution of human preference rankings.

The reviewers identified the importance of the problem, the statistically-grounded framework, extensive evaluation, broad applicability, clarity of presentation, and the novelty of the methodology all as strengths of the paper.

To further improve the paper, the authors should consider testing their approach on a wider range of benchmarks, tightening up their theoretical justifications, and so on. The reviewers left a number of potentially interesting suggestions that the authors should carefully consider.